# Nonlinear modeling of oral glucose tolerance test response to evaluate associations with aging outcomes

Grant Schumock[1]*, Karen Bandeen-Roche[1], Chee W. Chia[2], Rita R. Kalyani[3], Luigi Ferrucci[4], Ravi Varadhan[5]

1 Department of Biostatistics, Johns Hopkins University Bloomberg School of Public Health, Baltimore, Maryland, United States of America, 2 Clinical Research Unit, National Institute on Aging, National Institutes of Health, Baltimore, Maryland, United States of America, 3 Division of Endocrinology, Diabetes, and Metabolism, Johns Hopkins University School of Medicine, Baltimore, Maryland, United States of America, 4 Translational Gerontology Branch, National Institute on Aging, National Institutes of Health, Bethesda, Maryland, United States of America, 5 Department of Oncology, Johns Hopkins University School of Medicine, Baltimore, Maryland, United States of America

* gschumo1@jhmi.edu

**Data Availability Statement:** This paper relies on data from the Baltimore Longitudinal Study of Aging (BLSA), which are not publicly available due to the consent form language that the BLSA

## Abstract

As people age, their ability to maintain homeostasis in response to stressors diminishes. Physical frailty, a syndrome characterized by loss of resilience to stressors, is thought to emerge due to dysregulation of and breakdowns in communication among key physiological systems. Dynamical systems modeling of these physiological systems aims to model the underlying processes that govern response to stressors. We hypothesize that dynamical systems model summaries are predictive of age-related declines in health and function. In this study, we analyze data obtained during 75-gram oral-glucose tolerance tests (OGTT) on 1,120 adults older than 50 years of age from the Baltimore Longitudinal Study on Aging. We adopt a two-stage modeling approach. First, we fit OGTT curves with the Ackerman model—a nonlinear, parametric model of the glucose-insulin system—and with functional principal components analysis. We then fit linear and Cox proportional hazards models to evaluate whether usual gait speed and survival are associated with the stage-one model summaries. We also develop recommendations for identifying inadequately-fitting nonlinear model fits in a cohort setting with numerous heterogeneous response curves. These recommendations include: (1) defining a constrained parameter space that ensures biologically plausible model fits, (2) evaluating the relative discrepancy between predicted and observed responses of biological interest, and (3) identifying model fits that have notably poor model fit summary measures, such as $R^2_{\text{pseudo}}$, relative to other fits in the cohort. The Ackerman model was unable to adequately fit 36% of the OGTT curves. The stage-two regression analyses found no associations between Ackerman model summaries and usual gait speed, nor with survival. The second functional principal component score was associated with faster gait speed (p<0.01) and improved survival (p<0.01).

participants agreed to. The National Institutes of Health IRB supervises the BLSA and must approve any data release. To request data from the BLSA, please visit the BLSA website [https://www.blsa.nih.gov]/ and fill out a form. The BLSA Data Sharing Proposal Review Committee, which evaluates and approves all data requests/releases, meets once a month.

**Funding:** GS received funding from the National Institutes of Health, National Institute on Aging grant 5T32AG000247. KBR and RV received funding from the National Institutes of Health, National Institute on Aging grant 1UH3AG056933. CWC and LF received funding from the Intramural Research Program of the National Institutes of Health, National Institute on Aging. https://www.nia.nih.gov/. The content is solely the responsibility of the authors and does not necessarily represent the official views of the National Institutes of Health. The funders had no role in study design, data collection and analysis, decision to publish, or preparation of the manuscript.

**Competing interests:** The authors have declared that no competing interests exist.

## Introduction

Homeostasis relies on a complex network of connections between functioning physiological systems [1]. In good health, physiological systems are resilient—they can withstand and recover from stressors. Throughout the adult lifespan, these systems often exhibit loss of resilience and decline in performance [2]. In particular, loss of resilience in the physiological systems that govern stress response, the autonomic nervous system, energy regulation and production, and musculoskeletal integrity are hypothesized to contribute to biological aging and physical frailty [3, 4]. It is further hypothesized that loss of resilience is observable before physical frailty manifests. Because a physiological system must undergo stress to demonstrate resilience or lack thereof, tests that induce a stress response are well-suited for studying these systems. Dynamic stimulation tests (also known as "provocative tests") are in-clinic tests designed to stress specific physiological systems. The use of dynamic stimulation tests to evaluate resilience of physiological systems appears to hold promise for identifying older adults on adverse aging trajectories or at risk of incident frailty.

Varadhan et al. 2008 developed a framework for studying physiological systems by dynamical systems modeling [5]. This framework proposes parametrically modeling physiological response during a dynamic stimulation test with the idea that parameter estimates can provide insights into physiological functioning. The hypothesis that physiological response to dynamic stimuli varies by frailty status has been explored in the metabolic, stress response, and musculoskeletal systems [6]. One study examined glucose metabolism through an oral glucose tolerance test (OGTT) in a population of older women without diabetes. It found frail women—those demonstrating decreased resilience to stressors, as defined by the Fried frailty index—had significantly higher glucose levels two hours after glucose ingestion than non-frail women, yet baseline glucose levels were comparable [3, 7]. Another study examined the stress-response system in older women not taking corticosteroids through an adrenocorticotropic hormone stimulation test. Post-stimulation dehydroepiandrosterone levels were more rapidly elevated with increasing frailty, although mean pre- and post-stimulation levels did not significantly vary by frailty status [8]. A third study in older women found that prefrail and frail women had slower phosphocreatine recovery than nonfrail women after performing a 30-second maximal isometric contraction of the tibialis anterior muscle [9].

While studies have demonstrated frailty is associated with dysregulated physiological response to stressors, none have employed the mathematical modeling approach outlined in Varadhan et al. 2008. The supposed advantage of the mathematical modeling approach over simple nonparametric summaries is that parameters related to the rate of recovery from stimulus are hypothesized to summarize mechanisms governing physiological fitness or dysregulation [5]. We applied the framework laid out in Varadhan et al. 2008 to energy regulation and metabolism, the glucose-insulin system specifically.

We chose to study the glucose-insulin system because of its connections to several physiological systems linked to frailty including energy production and regulation, the musculoskeletal system, and the endocrine system [6]. Because of its connections to these key systems, functioning of the glucose-insulin system is associated with multiple adverse outcomes. Elevated fasting glucose levels have been linked to increased risk of cardiovascular disease, death, and stroke in older adults [10–12]. 2-hour plasma glucose (2hPG), measured during OGTTs, is an independent predictor of mortality [11, 13–16]. Additionally, in non-diabetic women, elevated 2hPG has shown an association with increased frailty status while fasting plasma glucose did not [7].

Various mathematical models have been proposed to describe the metabolism of glucose following the oral ingestion of a specified amount of glucose [17–21]. Some consider only

first-order interactions between glucose and insulin; others incorporate the effects of incretins and model the rate of glucose absorption from the intestines [17–20]. The Ackerman model is one of the simplest, but still widely-used, models [21]. The Ackerman model assumes blood glucose and insulin interact through a set of first-order linear differential equations and that the rate of glucose uptake from the intestines peaks immediately and proceeds to fall off slowly. While the Ackerman model does not model the underlying biological processes as explicitly as some alternative models, it was selected for this study because it only requires estimation of four parameters—one which estimates fasting glucose, another which controls the rate of an exponential decay function, one which controls the sinusoidal pattern of the glucose curve, and the last influences the amplitudes of the glucose curve oscillations. Although mathematical models have not been used in physiology and clinical medicine as an established methodology for explanatory or diagnostic purposes, they have been used extensively as investigative tools in metabolic and endocrine studies [22]. Here we propose to explore whether the Ackerman model parameters provide insights into age-related declines in function.

In applying the framework described in Varadhan et al. 2008, it became clear that fitting a mathematical model of the glucose-insulin system in a cohort of heterogeneous adults presents several challenges. We found significant variability in the shapes of OGTT curves between individuals. While some prior studies have categorized OGTT curves into "monophasic", "biphasic", or "continuous rise" shapes [23–26], we found a greater diversity of curve shapes in our data. The Ackerman model struggled to fit these atypical OGTT curves. This paper communicates the challenges we encountered in fitting dynamical systems data in a heterogeneous cohort of late middle aged and older adults and reports approaches we developed to address them. It aims to inform and guide researchers pursuing similar lines of inquiry in best analytic practices with their analyses of nonlinear dynamical systems models.

The rest of the paper is organized as follows. First, the data used in the case study are described. Next, the Ackerman model, a parametric, mathematical model of the glucose-insulin system, is introduced, as well as functional principal components analysis (fPCA), a non-parametric modeling approach. To complete the methods section, the regression analyses used in the case study are described. The results section explores the patterns of OGTT curves which proved difficult to mathematically model. We termed the Ackerman model fits associated with these curves "inadequate fits" and we present the criteria we devised to identify these fits in the results section. Findings from the fPCA analyses are also presented. To explore the utility of summary measures of glucose response dynamics for identifying older adults at risk for adverse outcomes, regression analyses of gait speed and mortality on these summary measures are presented. Lastly, we discuss possible causes of the difficulties with fitting the Ackerman model, make suggestions for other researchers seeking to implement nonlinear model fitting to a heterogeneous cohort, and identify future work needed.

## Methods

### Data source

The data for this case study comes from the Baltimore Longitudinal Study of Aging (BLSA), the longest-running US study on aging [27]. The BLSA is a community-based cohort study that continuously enrolls healthy volunteers aged 20 years and older who live within two hours driving from Baltimore, Maryland [27]. The BLSA follows participants for life. The study was designed to answer mechanistic questions about aging and the transition from health to disease with age. Participants undergo extensive, 3-day testing every 1–4 years, with older participants visiting more frequently. Data used in this manuscript are from the BLSA which is approved by the Institutional Review Board of the National Institute of Environmental Health

Sciences under protocol 03-AG-0325. BLSA participants given written informed consent at each study visit. The National Institute of Aging provided the data used in this manuscript to the study team upon review of the study plan and completion of a data use agreement. The data provided by the BLSA to the study team were retrospective and anonymized and thus were acknowledged to not be human subjects research by the study team's IRB—the Johns Hopkins Medicine IRB. Participants aged 50 years and older who had an OGTT conducted between January 1, 2001, and March 11, 2020, were included in the provided data set. BLSA participants with a history of diabetes and on insulin therapy do not undergo OGTTs. The data were provided to the researchers on August 18, 2020.

## Measures

The OGTT procedure consists of an overnight fast of at least 10 hours, consumption of an oral 75 g glucose load, and blood draws at 0, 20, 40, 60, 80, 100, and 120 minutes post-ingestion [28]. Participants were prohibited from smoking, eating, or exercising during the test [29]. Plasma glucose concentration was measured using a glucose oxidase analyzer (Abbott, Chicago, IL 2001–2006; Beckman Instruments, Brea, CA, 2006–2009; YSI Incorporated, Yellow Springs, OH, 2009 onwards) [28, 30].

Regression analyses employed usual gait speed and mortality as outcomes. Usual gait speed is measured at BLSA visits by asking participants to walk at their "usual, comfortable pace" over 6 meters of an uncarpeted floor [31]. Two trials are performed, and the faster of the two is used in our analyses. Death dates are tracked for BLSA participants, including those who drop out, by notice by family members, participant's physician of record, and search of the National Death Index. Participants who had not died by May 2, 2020, were considered censored at that date.

Because participants often undergo OGTTs and usual gait speed tests at multiple visits, we used the first visit in which both tests were completed for each participant. We also obtained personal characteristic variables including age at the recorded visit, BMI, self-reported sex, race, and smoking history. For the survival models, time since this visit was used.

The study population included 783 White, 276 Black, 24 non-Chinese/Japanese/Filipino Asian or other Pacific Islander, 11 other non-White, 11 Chinese, 6 not classifiable, 5 Filipino, 4 American Indian or Alaskan Native, and 4 Japanese participants. Due to small sample sizes, races were recoded into White, Black, and Non-White/Non-Black. The Non-White/Non-Black race category has a small number of participants and is highly heterogeneous. The group is only defined to allow modeling differences between White and Black participants—model coefficients for the Non-White/Non-Black category will not be interpreted.

## Data analysis

We separated analyses into three categories, stage-one, stage-two, and auxiliary models. Stage-one models were used to provide individual-level summaries of OGTT curves. A parametric approach, the Ackerman model, and a nonparametric approach, fPCA, were employed as stage-one models. Stage-two models were used to assess associations between the outputs of stage-one models and the outcomes of interest, usual gait speed and risk of death, while controlling for personal characteristics. Auxiliary models explored relationships between stage-one model outputs and personal characteristics. Data from seven participants were included in the stage-one models but excluded from the stage-two and auxiliary models due to missing smoking history status.

**Ackerman model.** The Ackerman model is a lumped-parameter model of blood glucose during an OGTT [21]. It was developed to test the hypothesis that physiological rhythms may

differentiate between health and disease. Ackerman et al. created this model by representing the glucose-insulin system in a block diagram including glucose absorption from the digestive system and liver; removal of glucose by the kidneys, liver, and other tissues; insulin release from the pancreas; insulin destruction; and interactions between insulin and glucose affecting these rates [21]. The differential equations governing the Ackerman model are

$$\dot{H} = -l_1 H + l_2 + l_3 Y \tag{1}$$

and

$$\dot{Y} = -l_4 Y + l_5 - l_6 H + I \tag{2}$$

where $Y$ is blood glucose concentration, $H$ is insulin concentration, $\dot{Y}$ and $\dot{H}$ are the rates of change in blood glucose and insulin concentrations, $I$ is the rate of change in blood glucose due to absorption from the intestines, $l_1 H$ is the average rate of insulin removal independent of glucose, $l_2$ is the average rate of release of insulin from the pancreas independent of glucose, $l_3 Y$ is the net increase in the rate of release of insulin due to glucose, $l_4 Y$ is the average rate of glucose removal independent of insulin, $l_5$ is the average rate of release of glucose into the blood, and $l_6 H$ is the net increase in the average rate of glucose removal from the blood due to insulin [21]. Applying initial conditions and the assumption that $\dot{I}$ follows a Dirac delta function centered at the start of the test yields the integral form of the Ackerman model

$$Y(t) = Y_F + A e^{-kt} \sin(\omega t) \tag{3}$$

where $Y(t)$ is the blood glucose concentration at time $t$, $Y_F$ is the fasting blood glucose concentration, $A$ is a scale parameter, $k$ is an exponential decay parameter, and $\omega$ is a sinusoidal parameter. For complete details, see Ackerman et al. 1964.

Allowing for differences in the parameters between individuals and measurement or model error gives

$$Y_i(t) = Y_{F_i} + A_i e^{-k_i t} \sin(\omega_i t) + \epsilon_{i,t} \tag{4}$$

where $i$ denotes the individual, $Y_i(t)$ is the measured plasma glucose concentration for individual $i$ at time $t$, and $\epsilon_{i,t}$ is the error between the measured and the Ackerman model's theoretical plasma glucose concentration for individual $i$ at time $t$. Plasma glucose is used in lieu of whole blood glucose because plasma glucose was measured in the BLSA and the two are approximately proportionally equivalent.

Nonlinear least squares was used to fit the Ackerman model to individuals' OGTT curves. The parameter estimates $\hat{Y}_{F_i}, \hat{A}_i, \hat{k}_i, \hat{\omega}_i$ were found such that

$$\sum_{t \in \{0, 20, \dots, 120\}} \left( Y_i(t) - Y_{F_i} - A_i e^{-k_i t} \sin(\omega_i t) \right)^2 \tag{5}$$

was minimized at $(\hat{Y}_{F_i}, \hat{A}_i, \hat{k}_i, \hat{\omega}_i)$. Homoscedastic errors with mean 0 were assumed.

Ackerman et al. were interested in modeling the glucose-insulin system because they believed that natural periods of physiological systems could distinguish disease from health. The disease and periodicity combination they investigated was diabetes and a parameter they termed "effective period". Effective period is a measure of the time it takes for the glucose-

insulin system to recover from the glucose stimulus during an OGTT and is given by

$$T_{eff_i} = \frac{2\pi}{\sqrt{\omega_i^2 + k_i^2}}$$ (6)

[21].

Large effective periods indicate slow recovery to the glucose stimulus, while low effective periods indicate rapid recovery of the glucose-insulin system. Ackerman et al. used a cutoff value of 4 hours and found it performed well for differentiating between persons with versus without diabetes. While the aim of the present paper is unrelated to diabetes diagnosis, effective period may be a concise measure of resilience of the glucose-insulin system during an OGTT.

*Parameter bounds*. For the Ackerman model, bounds must be placed on the parameters so that the curve is biologically plausible. For example, a negative $Y_{F_i}$ would correspond to a negative basal glucose concentration; a negative $A_i$ with a positive $\omega_i$, or vice versa, would signal an initial decline in glucose concentrations after consumption of the glucose load; a negative $k_i$ would indicate glucose concentrations oscillate with increasing amplitude in time. Upper bounds are also necessary for biological plausibility as well as to achieve algorithm convergence with a reasonably high likelihood. However, cutoff points are not as clear as for the lower bounds. For example, extremely large values of $\omega_i$ correspond to near-instantaneous oscillations in plasma glucose concentration. Likewise, very large values of $k_i$ correspond to near-instantaneous return to basal glucose levels. The parameter bounds we implemented were $Y_{F_i} \in [0, 2Y_i(0)]$, $A_i \in [0, 20(\max_{t \in \{0,20,\ldots,120\}} Y_i(t) - Y_i(0))]$, $k_i \in [0, 10]$, $\omega_i \in [0, 2\pi]$, where the units for these parameters are mg/mL for $Y_{F_i}$ and $A_i$, and 1/hour for $k_i$ and $\omega_i$. These values were chosen through a combination of reasoning and trial-and-error aimed at achieving convergence in the estimation algorithm.

The lower bounds for these parameters were chosen based solely on physiology. An added benefit to imposing lower bounds of 0 on the parameters is the model is structurally identifiable except for $A_i = 0$. We did not find non-identifiability of $A_i = 0$ to be an issue in practice, as this corresponds to a flat OGTT curve—none of which we observed.

*Challenges due to non-linearity*. The Ackerman model can be difficult to fit to OGTT data due to the model's inherent nonlinearity. Glucose concentrations and time measurements were converted into mg/ml and hours, respectively. These units were chosen to scale the parameters such that the optimal parameter estimates lie between 0 and 10 for most OGTT curves. Rescaling has been shown to improve algorithms' abilities to find optimal parameter sets—especially when the parameters are of different orders of magnitude on the originally scaled data [32].

Another issue that can arise when fitting the Ackerman model is an inability to converge to the global minimum. Because the model is nonlinear, the least squares equation may have several local minima. Optimization routines are not guaranteed to find the global minimum, especially when initial points are chosen poorly [33].

*Fitting algorithm*. The Ackerman model was fit to individuals' OGTT curves using the *nls* function in the stats R package v4.2.1. The port algorithm for nonlinear regression was used in the *nls* function. An initial value of 1 was chosen for all parameters. When the algorithm did not converge to a satisfactory fit, the process was restarted with an initial value chosen from a multivariate uniform distribution using the parameter bounds as the bounds of the distribution. This was repeated until a satisfactory fit was found, or 1,000 initial conditions were tried. Satisfactory fits are defined under the subsection "A Taxonomy of Responses" in the Results section.

## Functional principal components analysis

As nonlinear, parametric models pose model-fitting challenges, nonparametric methods were also explored. Because the shape of the glucose curve is hypothesized to be related to resilience of the glucose-insulin system to a glucose load, a method that captures this information was desired. Functional principal components analysis was determined to be an appropriate method. This method was also recommended by Varadhan et al. 2008 as an alternative to parametric modeling for systems where a mathematical model of the dynamics is not available. fPCA identifies eigenfunctions that explain the dominant modes of variation in functional data around a mean curve. For OGTT data, fPCA can be used to identify dominant patterns in plasma glucose response to an oral glucose load. Past research has employed fPCA on OGTT data and found principal component (PC) scores better discriminate between pregnant women who went on to develop gestational diabetes and those who did not when compared to traditional OGTT summary measures [34]. Using fPCA to explore associations of OGTT responses with gait speed and mortality is novel.

The first step of fPCA is fitting smoothing curves to individuals' OGTT measurements. This can be done in many ways, smoothing splines and kernel smoothing being among the most popular [35, 36]. We used 6 cubic B-splines, with a roughness penalty of $\lambda = 733$ on the second derivative of the smoothed curves. This value of $\lambda$ was chosen using the generalized cross-validation criterion described in section 5.4.3 of Ramsay et al. 2005. For smooth curves $x_1(t), \ldots, x_n(t)$, the mean curve is calculated as

$$\bar{x}(t) = \frac{1}{n}\sum_{i=1}^{n} x_i(t). \tag{7}$$

Mean-centering each curve, principal component functions are calculated as

$$\varphi_j^*(t) = \arg\max_{\varphi_j(t)} \frac{1}{n}\sum_{i=1}^{n}\left(\int_\tau \phi_j(t)[x_i(t) - \bar{x}(t)]\right)^2 dt \tag{8}$$

where $\phi_j(t)$ are orthonormal functions, i.e. $\int_\tau \phi_j(t)^2 dt = 1$, $\int_\tau \phi_j(t)\phi_k(t) = 0$ for $j \neq k$, and $\sum_{i=1}^{n}\left(\int_\tau \phi_1(t)[x_i(t) - \bar{x}(t)]\right)^2 dt > \sum_{i=1}^{n}\left(\int_\tau \phi_2(t)[x_i(t) - \bar{x}(t)]\right)^2 dt > \ldots$

Functional principal component scores are calculated as

$$\eta_{i,j} = \int_\tau \phi_j^*(t)[x_i(t) - \bar{x}(t)]dt. \tag{9}$$

Predicted curves using the first $k$ eigenfunctions are calculated as

$$\hat{x}_i(t) = \bar{x}(t) + \sum_{j=1}^{k} \eta_{i,j}(t)\phi_j^*(t). \tag{10}$$

The end products of fPCA are a mean curve which is an average of all of the OGTT curves, principal component functions which explain the maximal variation in OGTT curves about the mean, and functional principal component scores which explain how an individual's OGTT curve varies about the mean. While fPCA may not provide deeper insights into the mechanistic parameters, it has a few distinct advantages over a parametric modeling approach. Firstly, fPCA is nonparametric so it does not assume a potentially incorrect model about the data-generating distribution—a major cause of convergence issues in the parametric modeling setting. Secondly, fPC scores are uncorrelated by design so there are no potential concerns of

multicollinearity when using these scores in regression models. The R package fda v6.0.5 was used to perform the functional principal components analysis.

## Stage-two outcome models and auxiliary analyses

Stage-two models were used to explore whether the summaries produced by the stage-one models are associated with the outcomes of interest, usual gait speed and risk of death. This was done to test the hypothesis that our stage-one models are useful for summarizing OGTT curves in a way that provides information about whether individuals are on adverse aging trajectories. Linear regression models were used to examine the relationships between usual gait speed and stage-one model summaries. Cox proportional hazards models were used to model the risk of death as a function of stage-one model summaries. For the stage-two models, the estimated Ackerman model parameters and fPC scores were standardized. The estimated Ackerman model parameters were standardized by subtracting the corresponding mean amongst the adequate Ackerman model fits and then dividing by the standard deviation. The fPC scores were standardized by only dividing by the standard deviation of the corresponding fPC score amongst all fits since the fPC scores by formulation have mean 0. The estimated Ackerman model parameters from inadequate fits in the stage-one models were not included in the stage-two models. This was prevented by including an indicator variable, $\mathbb{1}$(Adequate Fit) which was 0 for inadequate fits and 1 for adequate fits. In the regression tables, $\mathbb{1}$(Adequate Fit) indicates a separate intercept between adequate and inadequate fits, and $\hat{T}_{\text{eff}} \cdot \mathbb{1}$(Adequate Fit) indicates the effect of a 1 hour increase in $\hat{T}_{\text{eff}}$ amongst the adequate fits.

Auxiliary models were used to explore how stage-one model summaries were related to personal characteristics. Logistic regression was used to model the odds that the Ackerman model inadequately fit an individual's OGTT curve based on personal characteristics. Linear regression was used to model the associations between estimated effective period and personal characteristics, excluding individuals for whom the Ackerman model provided an inadequate fit. Linear regressions were also used to model the associations between fPC scores and personal characteristics.

## Results

### Ackerman model fits

**A taxonomy of responses.** The Ackerman model fitting procedure described in "Model Fitting Procedure" of Methods produced Ackerman model fits for every OGTT curve except one. Upon further inspection, the one OGTT curve that the fitting algorithm failed to model showed highly atypical behavior. The glucose concentration of this individual decreased following the consumption of the glucose load rather than increasing as expected. This type of behavior is difficult to explain biologically. The Ackerman model is not equipped to model this phenomenon, and it is not of interest in this paper's analyses. Thus, OGTT curves with lower 20 minute glucose concentrations than baseline concentrations were excluded (n = 4). This resulted in 1,120 individuals included in the subsequent analyses.

Although the fitting algorithm converged for almost every OGTT curve, not every fit appeared to fit the data well. In general, three patterns of inadequate fits emerged: fits with estimated parameters on the boundaries of the allowed parameter space, fits that poorly predicted the peak glucose concentration, and fits with relatively low $R^2_{\text{pseudo}}$. Adequate fits were defined as Ackerman model fits which were not classified as inadequate fits. Classifying fits as adequate or inadequate was done so that unreliable parameter estimates could be treated separately in the stage-two models.

Examples of predicted OGTT curves under the Ackerman model are shown in Fig 1. All of these model fits were deemed to be adequate fits to the observed data. Panels A-D show Ackerman models with increasing effective periods (1.9, 3.3, 4.9, and 8.4 hours, respectively).

*Boundary fits*. Fits with parameter estimates on the boundaries of the allowed parameter space were termed "boundary fits". Fig 2 panels A-D show example boundary fits. Panel A shows a model fit where $\hat{A}_i = 2 \cdot Y_i(0)$, panel B where $\hat{k}_i = 0$, panel C where $\hat{w}_i = 2\pi$, and panel D where $\hat{A}_i = 2\dot{Y}_i(0)$ and $\hat{k}_i = 0$.

As shown by these plots, boundary fits typically do not appear to model the observed data well. Boundary fits can occur for two reasons. First, there may exist an optimal Ackerman model for a particular OGTT curve, but one or more of the parameters of this optimal model lie outside of the allowed parameter space. In this case, the model-fitting algorithm returns fitted parameter estimates that lie near the unrestricted maximum likelihood estimate (MLE), but lie on the parameter space boundary. This appears to be the cause of boundary fits in which $\hat{k}_i$ and $\hat{\omega}_i$ are on the boundary of the parameter space.

When $\hat{A}_i$ is on the boundary of the parameter space, it appears to be caused by a nearly flat likelihood near the MLE. For these fits, the OGTT curves can be equally, or nearly equally, well-explained by multiple sets of Ackerman model parameters. Some authors call these "near redundant" or "near singular" fits [37, 38]. These fits are caused by practical non-identifiability. For these curves, the nonlinear model-fitting algorithm has difficulty finding an optimal fit without imposing upper bounds on $\hat{A}_i$. For example, the likelihood corresponding to the black points in Fig 2 is nearly flat. Multiple sets of parameters are nearly equally capable of fitting this curve. The OGTT curve in Fig 2 panel A can be fit nearly equally well by the Ackerman model with vastly different $\hat{A}_i$'s and $\hat{\omega}'_i s$: ranging from 10 to 200 mg/dL and 0.31 to 0.02 1/ hour, respectively. Even with this range of parameter estimates, these fits predict nearly indistinguishable OGTT curves with nearly identical residual sum of squares (489, 486, and 486

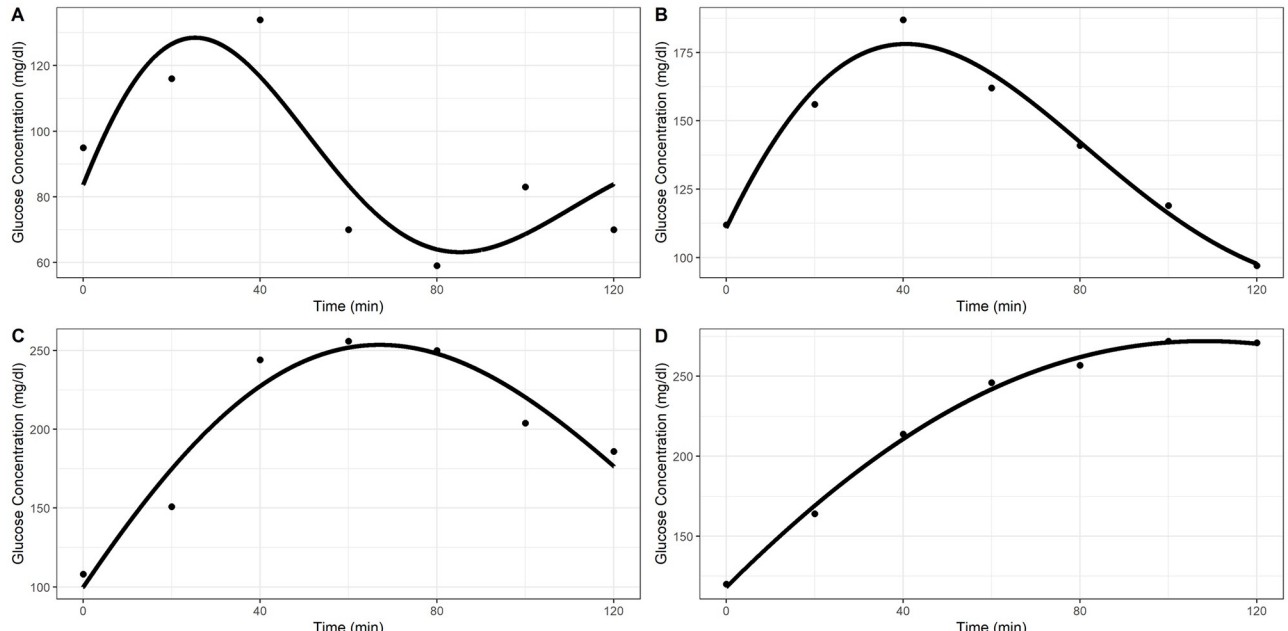

**Fig 1. Adequate Ackerman model fits.** Estimated effective periods for the model fits by panel: A—1.9 hours, B—3.3 hours, C—4.9 hours, D—8.4 hours.

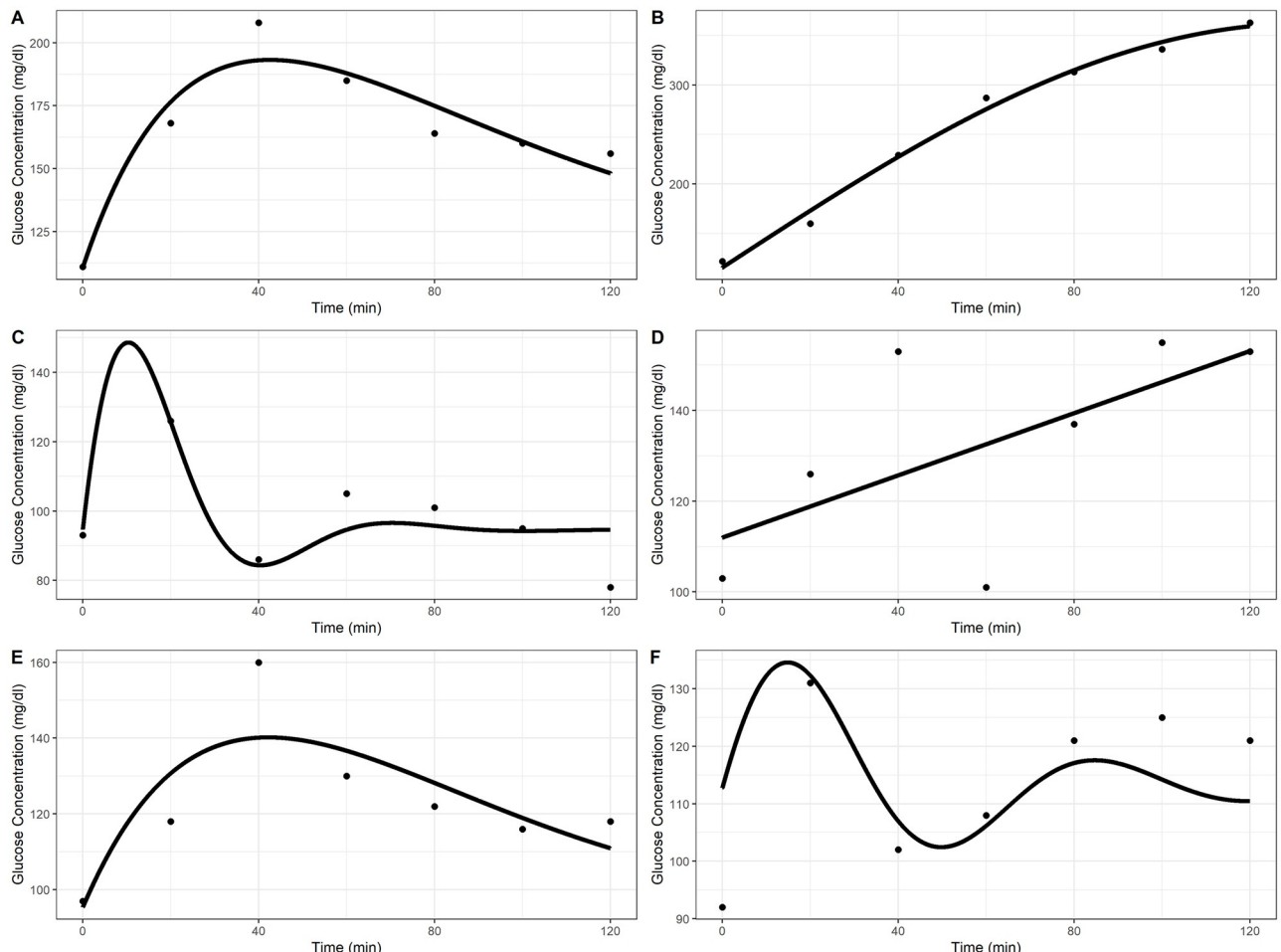

**Fig 2. Inadequate Ackerman model fits.** Reasons for inadequate fit classification by panel: A—$\hat{A}$ on boundary of parameter space, B—$\hat{k}$ on boundary of parameter space, C—$\hat{\omega}$ on boundary of parameter space, D—$\hat{A}$ and $\hat{k}$ on boundary of parameter space and small $R^2_{\text{pseudo}}$, E—large underestimation in max glucose concentration, F—small $R^2_{\text{pseudo}}$.

mg$^2$/dl$^2$, respectively). These fits can be obtained by changing the upper bound on $\hat{A}_i$. Each of these fits produced an $\hat{A}_i$ on the upper bound of the allowed parameter space. Since these fits are nearly indistinguishable yet lead to different parameter estimates, they are deemed inadequate fits. In a sensitivity analysis, we created profile likelihood plots for each OGTT to examine for practical non-identifiability of the Ackerman model. We found several instances of practical non-identifiability, as evidenced by flat profile likelihoods near profile maxima. However, these OGTT curves were found to be a subset of the inadequate fits we already identified, so these Ackerman model fits were already excluded from the stage-two models.

While boundary fits could be avoided by not restricting the parameter space, this exchanges one issue for many. For OGTT curves that correspond to an MLE in a biologically infeasible region, the resulting model fits well but fails to provide a realistic model of reality. For example, a $\hat{k}_i$ of −0.03 1/hour fits the OGTT curve in Fig 2 panel A well. However, this model predicts increasingly large oscillations in glucose concentration over time. For boundary fits where the likelihood is nearly flat at the MLE, removing the parameter space restrictions severely limits

the model-fitting algorithm's ability to converge. As described in the previous paragraph, increases in $\hat{A}_i$ can be counteracted by decreases in $\hat{\omega}_i$ to produce near-identical fits to some OGTT curves. Because the residual sum of squares of these fits are nearly identical, it is generally not feasible to find the MLE without imposing upper bounds on the parameter space.

*Extrapolated fits.* Another pattern of inadequate fits is characterized by predicted peak glucose concentration that differs significantly from observed peak glucose concentration. We have termed this pattern of fits "extrapolated fits", as the predicted peak glucose is often much greater than any observed glucose concentration. The Ackerman model fit in Fig 2 panel C shows an example of this type of fit. Mathematically, extrapolated fits are defined as fits where the maximum predicted glucose concentration occurs during the test and

$$\frac{|Y_{max,pred} - Y_{max,obs}|}{Y_{max,obs}} \geq \Delta_{\text{tol}} \tag{11}$$

where $Y_{max,pred}$ and $Y_{max,obs}$ are the predicted and observed maximum glucose concentrations and $\Delta_{\text{tol}}$ is the allowed tolerance. The first condition, that the maximum predicted glucose concentration occurs during the test, prevents well-fitting but slow-peaking fits from being categorized as extrapolated fits. The condition shown in 11 puts a restriction on the relative difference between the predicted and observed maximum glucose concentration. Based on our data, we set $\Delta_{\text{tol}}$ to be 10%. This cutoff in relative difference was decided upon by inspection of $Y_{max,pred}$ against $Y_{max,obs}$, shown in Fig 3, as well as visual inspection of OGTT curves with Ackerman model fits near 10% relative difference in predicted and observed peak glucose concentrations. Fig 3 also shows several Ackerman model fits underpredict the peak glucose concentration. The fit in Fig 2 panel E shows an example of this. Fits with large underprediction of the peak glucose concentration do not model the data well, so the absolute value signs in Eq 11 are necessary to ensure these fits are categorized as extrapolated fits.

It should be noted that an individual's peak glucose during an OGTT likely occurs in the time between two of the observed measurements. Thus, the observed peak glucose is almost certainly lower than the true peak glucose. Care should be taken to ensure that a screening criterion such as that presented in Eq 11 is not too stringent given that the true peak glucose is not measured. For this study, we did not find this to be an issue as the Ackerman model typically underpredicted the peak glucose when the predicted and observed peaks varied by more than 10% relative difference.

*Low pseudo-$R^2$.* The last class of inadequate fits relates to fits with low $R^2_{\text{pseudo}}$. Initially, we explored categorizing model fits by their absolute residual sum of squares. However, some model fits have high residual sum of squares while visually appearing to fit the data well. Fig 1 panel B shows an example where the Ackerman model appears to fit the OGTT curve well but has a large residual sum of squares. Further investigation revealed several cases where large residual sum of squares did not correspond to visually inadequate fits, and cases where the fit was clearly inadequate but did not produce a large residual sum of squares. Classifying fits as inadequate based on their pseudo-$R^2$ appears to match better with visual inspection. The pseudo-$R^2$ measure used is from [39] and is given by

$$R^2_{\text{pseudo}_i} = 1 - \frac{\text{Deviance}_i(Y_i, \hat{Y})}{\text{Deviance}_i(Y, \bar{Y})} = 1 - \frac{\sum_{t \in \tau}(Y_{i,t} - \hat{Y}_{i,t})^2}{\sum_{t \in \tau}(Y_{i,t} - \bar{Y}_i)^2} \tag{12}$$

Larger $R^2_{\text{pseudo}}$ indicate the Ackerman model explains more variability in the data relative to an intercept-only linear model. It is well-established that $R^2$ can be problematic for model selection of non-linear models [40]. However, in this research $R^2_{\text{pseudo}}$ is being used to classify

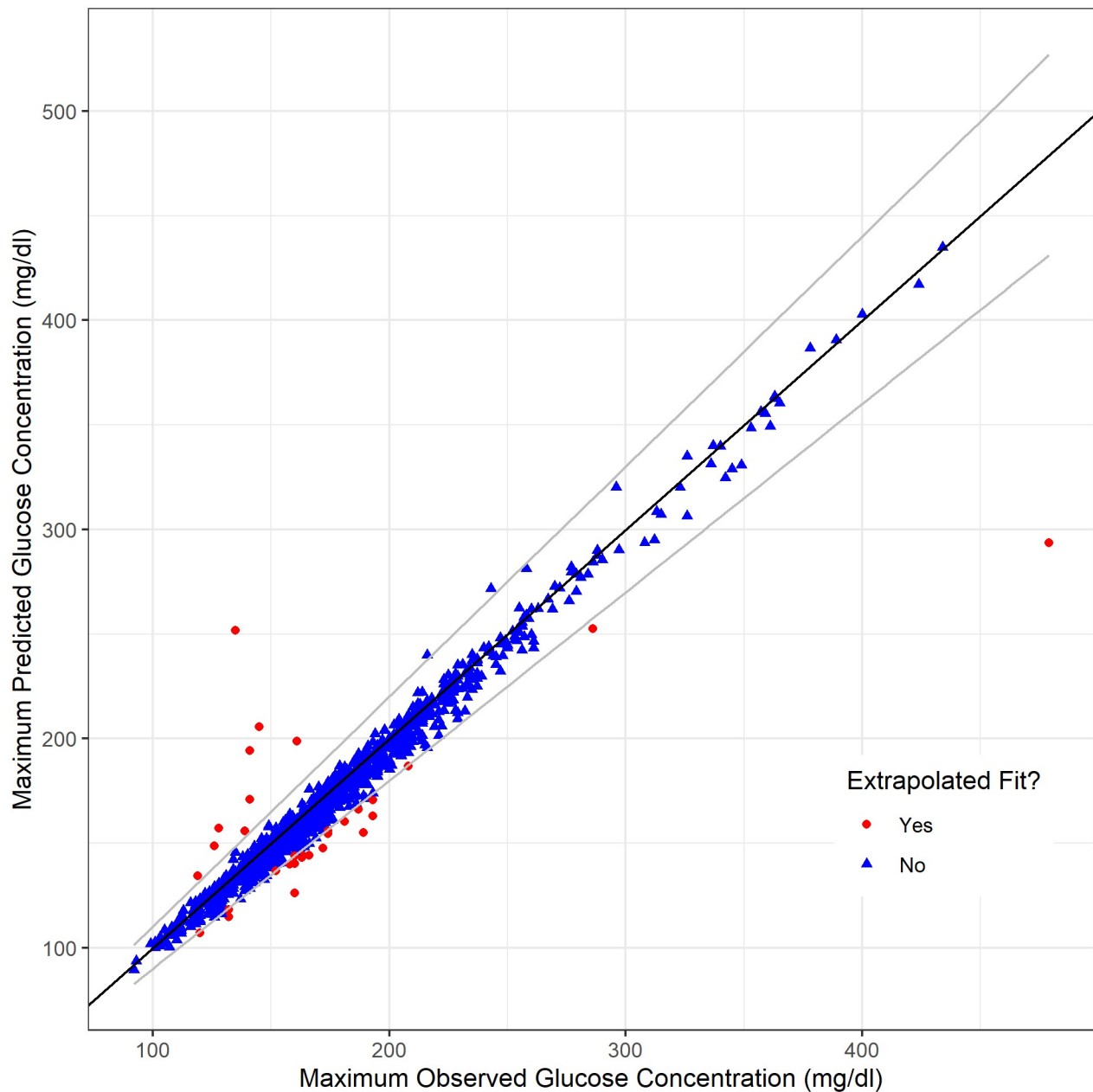

**Fig 3. Predicted and observed maximum glucose concentrations.** Black line corresponds to perfect prediction. Grey lines correspond to a 10% relative difference. Blue triangles outside of the grey lines indicate fits for which the predicted maximum glucose concentration occurred later than the end of the OGTT.

some Ackerman model fits as inadequate—no alternate nonlinear model is used. Furthermore, $R^2_{\text{pseudo}}$ is only one criterion that is used to evaluate whether the Ackerman model fit is inadequate or not.

The distribution of $R^2_{\text{pseudo}}$ for all Ackerman model fits and the non-boundary fits are shown in S1 Fig. As expected, the non-boundary fits in general have larger $R^2_{\text{pseudo}}$. Using S1 Fig as well as visual inspection of selected Ackerman model fits, a cutoff of $R^2_{\text{pseudo}} = 0.7$ was established.

Fits with $R^2_{\text{pseudo}}$ less than 0.7 are classified as inadequate fits. Fig 2 panel F shows an example of one such fit, with an $R^2_{\text{pseudo}} = 0.4$

*Inadequate fit screening results.* Applying the boundary fit, extrapolated fit, and pseudo-$R^2$ criteria to the Ackerman model fits of the BLSA OGTT data resulted in 717 (64%) adequate and 403 (36%) inadequate fits. Among all of the fits, 376 were boundary fits, 31 were extrapolated fits, and 84 were low $R^2_{\text{pseudo}}$ fits.

## Functional principal components analysis

Functional principal components analysis was applied to the same set of OGTT curves as the Ackerman model-fitting algorithm. 90.5%, 6.9%, and 2.0% of the variance in the smoothed OGTT curves was explained in the first, second, and third principal components, respectively. In Frøslie et al.'s research, they found that the third principal component provided necessary information about the shape of the OGTT curve [34]. Based on this, as well as the scree plot corresponding to our fPCA, we also retained the first three principal components. The scree plot, mean curve, and first three eigenfunctions are shown in Fig 4. The mean curve shows the average fasting glucose in the study was approximately 97 mg/dL, with a peak glucose concentration of about 161 mg/dL at 49 minutes, and a two-hour glucose concentration of 130 mg/dL.

The eigenfunctions show the principal modes of variation about the mean curve. Based on these eigenfunctions, large PC1 scores indicate reduced clearance of plasma glucose in the latter half of the OGTT relative to the average. Large PC2 scores correspond to earlier peak glucose and increased clearance after 75 minutes. Large PC3 scores indicate higher glucose concentrations at the start and end of the test, with reduced concentrations during the middle. Dysregulated glucose-insulin systems are associated with impaired glucose clearance. OGTT curves corresponding to high PC1 and low PC2 scores are likely indicative of dysregulation, and low PC1 with high PC2 scores may be indicative of above-average glucose-insulin system functioning. The third eigenfunction appears to follow the biphasic OGTT curve pattern that has been described in diabetes literature [23, 24]. Biphasic OGTT curves are thought to indicate healthy glucose regulation, and have been found to be associated with reduced risk of incident type 2 diabetes [25]. Thus, individuals with high PC3 scores may have above-average glucose-insulin system functioning. However, the third eigenfunction is also characterized by elevated predicted basal glucose. OGTT curves with high PC3 scores may be connected to impaired fasting glucose with normal glucose tolerance.

Unlike with the Ackerman model, we did not intentionally identify poor-fitting predicted OGTT curves fit using fPCA. First, the eigenfunctions do not have a direct biological interpretation like the Ackerman model parameters do. While certain combinations of PC scores could lead to predicted OGTT curves which are not biologically plausible, this cannot be determined by examining only one PC score in isolation. Second, each PC score is influenced by all of the other curves in the data set. The Ackerman model is fit independently to each individual's OGTT curve so isolating inadequate fits does not impact other fits.

## Stage-one models

The Ackerman model-fitting algorithm and fPCA both summarize the information of the OGTT curve into a few metrics. Fig 5 shows the relationships between the estimated Ackerman model parameters and the fPC scores for the subset of OGTT curves that the Ackerman model adequately fit. Note there is a nonzero correlation between the fPC scores on this subset of the OGTT curves because there is a relationship between the fPC scores and the ability of

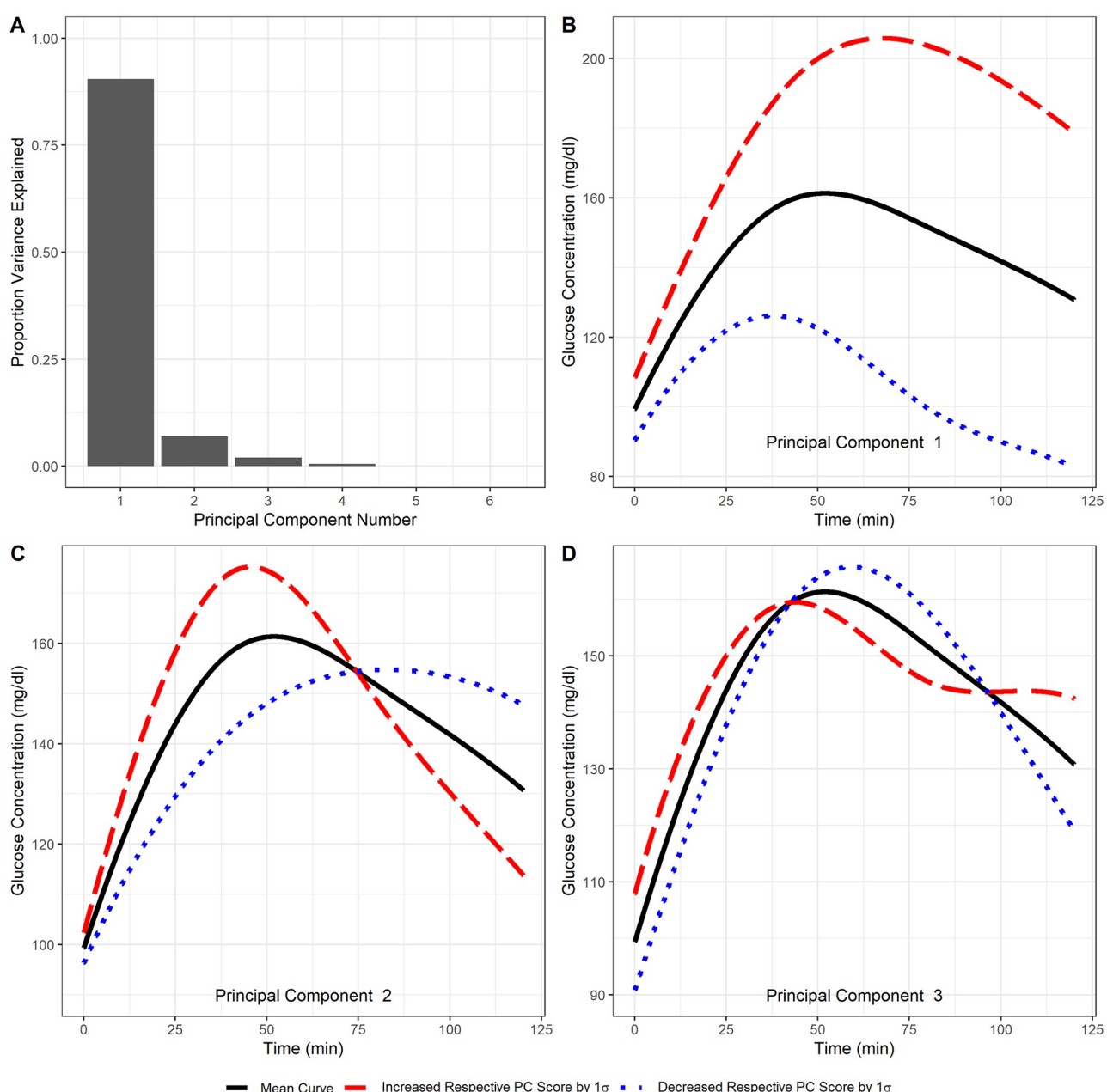

**Fig 4. Functional PCA plots.** Panel A—Proportion variance explained by principal component. Panels B-D—Eigenfunctions' deviations about the mean curve. The dashed red line shows a predicted OGTT curve for a 1 standard deviation increase in the respective fPC score with all other scores held at 0. The dotted blue line shows the same but for a 1 standard deviation decrease in the respective fPC score.

the Ackerman model to appropriately model the data. A logistic regression model of the odds of the Ackerman model fitting inadequately was constructed. The second and third PCs were found to strongly impact the odds of the Ackerman model fitting inadequately, with 1 standard deviation increases in PC1 and PC2 corresponding to odds ratios of 0.40 (95% CI:0.35—0.46, $P < 0.001$) and 2.01 (95% CI:1.62—2.51, $P < 0.001$), respectively. The effect of PC1 was more muted, with an odds ratio of 1.03 95% CI:(1.00, 1.06). Simply put, the Ackerman model was most capable of fitting OGTT curves which displayed a single, strong peak followed by a

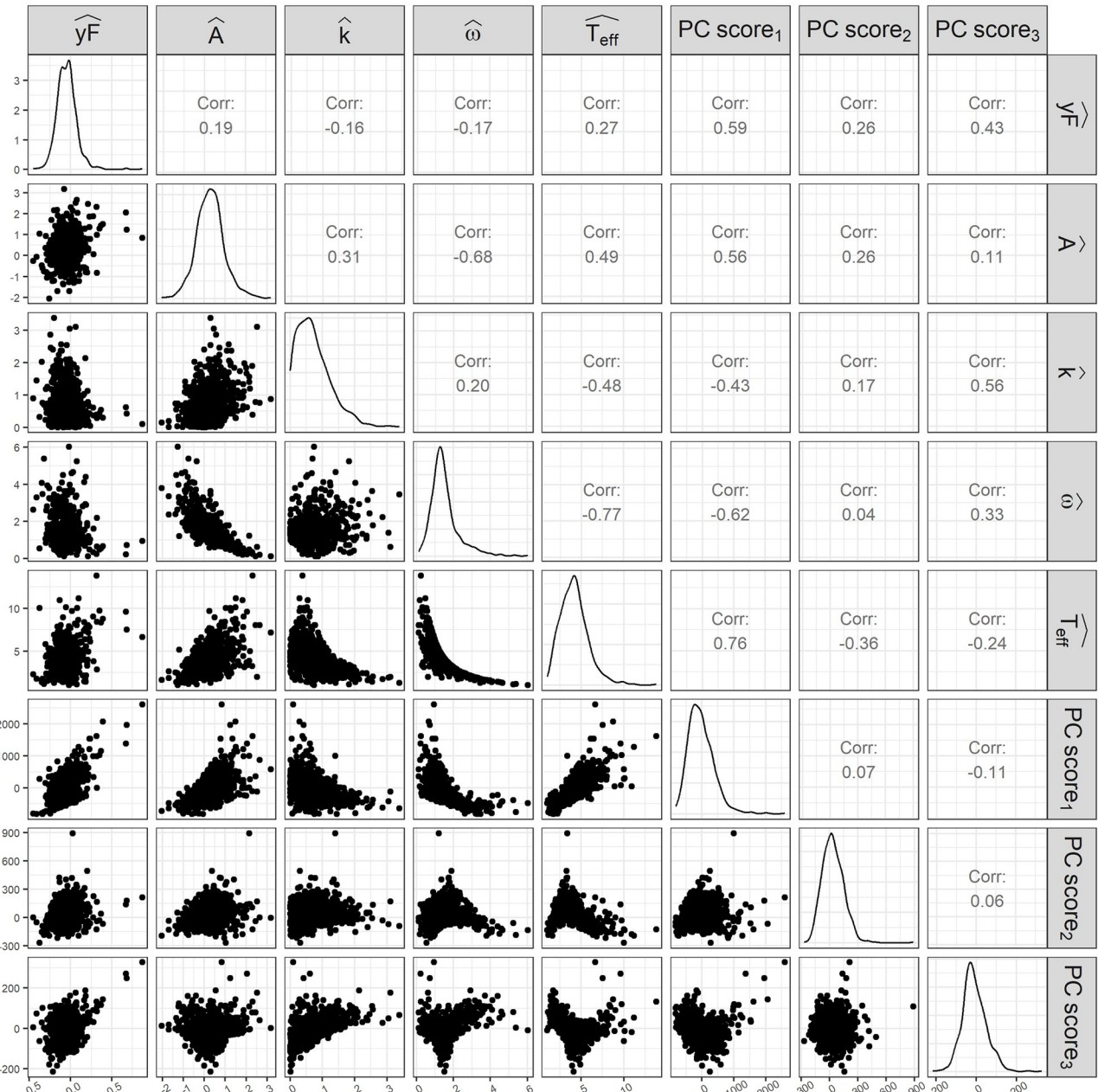

**Fig 5. Ackerman parameters and fPC scores scatterplot matrix.** Scatterplots and correlations between estimated Ackerman model parameters and fPC scores for OGTT curves which the Ackerman model adequately fit.

rapid drop in glucose levels. The Ackerman model struggled to fit OGTT curves with multiple peaks and curves which displayed slow declines in glucose levels after a peak was reached. A sensitivity analysis controlling for personal characteristics had little effect on the PCs' odds ratio estimates.

For the stage-two models, two Ackerman model parameters, $\hat{y}_F$ and $\hat{A}$, were logged to make their distributions more symmetric. Note the large positive correlations between PC1 and $\log(\hat{Y}_F)$, PC1 and $\log(\hat{A})$, PC1 and $\hat{T}_{\text{eff}}$, and PC3 and $\hat{k}$ shown in 5. There are also large

negative correlations between $\log(\hat{A})$ and $\hat{\omega}$, $\hat{T}_{\text{eff}}$ and $\hat{\omega}$, and PC1 and $\hat{\omega}$. These relationships reveal that PC1 is most closely related to the estimated Ackerman model parameters. The directions of these correlations are a result of the first eigenfunction's heightened glucose concentration, particularly at later time points, with increased $y_F$, $A$, and $T_{\text{eff}}$ indicating elevated glucose concentration and increased $k$ and $\omega$ indicating lowered glucose concentration at later time points. Interestingly, PC2 is not strongly correlated with any of the estimated Ackerman model parameters, although there is a weak negative correlation with $\hat{T}_{\text{eff}}$. This is to be expected as the second eigenfunction is characterized by heightened glucose concentration followed by reduced glucose concentration, and OGTT curves with smaller effective periods exhibit similar behavior. The relationship between PC3 and $\hat{k}$ is surprising because the third eigenfunction takes on a bimodal form that the Ackerman model cannot capture. S2 Fig includes data from all of the OGTT curves. Many of the relationships between the estimated Ackerman model parameters and each other or with the PCs are obscured by the presence of large outliers in $\hat{k}$, $\hat{\omega}$, and $\hat{T}_{\text{eff}}$. Additionally, the distributions of $\hat{A}$, $\hat{k}$, and $\hat{\omega}$ among the inadequately-fitting Ackerman models are bimodal due to congregations at the parameter boundaries. These distributions highlight the issues with the parameter estimates of boundary fits.

Fig 6 shows four OGTT curves, each with predicted Ackerman model and fPCA curves. For the OGTT curve in panel A of Fig 6, the Ackerman model provides an adequate fit; in panel B, a boundary fit with a small $R^2_{\text{pseudo}}$; in panel C, a boundary fit with a large $R^2_{\text{pseudo}}$; in panel D, an extrapolated fit. For curves in which the Ackerman model fit has low $R^2_{\text{pseudo}}$, the fPCA predictions are similar to the Ackerman model, albeit with greater curvature. The greater curvature flexibility of the fPCA fits allows the predicted maximum glucose concentration to better track the observed maximal glucose for typical OGTT curves. For abnormal OGTT curves like those in panels B and D, the predicted curves from the Ackerman model and fPCA differ markedly with neither fitting the data well. The OGTT curve in panel B shows 20 mg/dl oscillations in glucose concentration every 20 minutes, with even larger changes between 0 to 20 minutes and 100 to 120 minutes. The OGTT curve in panel D shows a glucose concentration at 20 minutes that is much greater than at any other time point. These curves display atypical glucose dynamics, so it is unsurprising that neither method is able to accurately model them.

In general, the Ackerman model more closely fits the observed OGTT data than the fPCA predictions. The average residual sum of squares for the Ackerman fits is 434 $\text{mg}^2/\text{dl}^2$ for the complete data set and 295 $\text{mg}^2/\text{dl}^2$ excluding the inadequate fits. The corresponding values for the fPCA predicted curves are 533 $\text{mg}^2/\text{dl}^2$ and 497 $\text{mg}^2/\text{dl}^2$. Better-fitting fPCA curves could be obtained by retaining more principal components in the predictions, using a smaller tuning parameter, or including more basis functions when smoothing the OGTT curves. However, the trade-offs with these include reduced interpretability and overfitting.

S3 Fig shows the residuals from the predicted OGTT curves based on the Ackerman model and fPCA fits. In general, the residuals are centered near 0 at each time point, so it does not appear that either method consistently overestimates or underestimates the glucose concentration over time. The Ackerman model appears to outperform fPCA at 0 and 120 minutes in particular.

## Outcome models and auxiliary analyses

Table 1 shows the relationship between demographic variables and the odds that the Ackerman model inadequately fits an individual's OGTT curve. The odds the Ackerman model provides an inadequate fit are 16% (95% CI: 2%—31%, $P = 0.020$) higher for a 10 year increase in

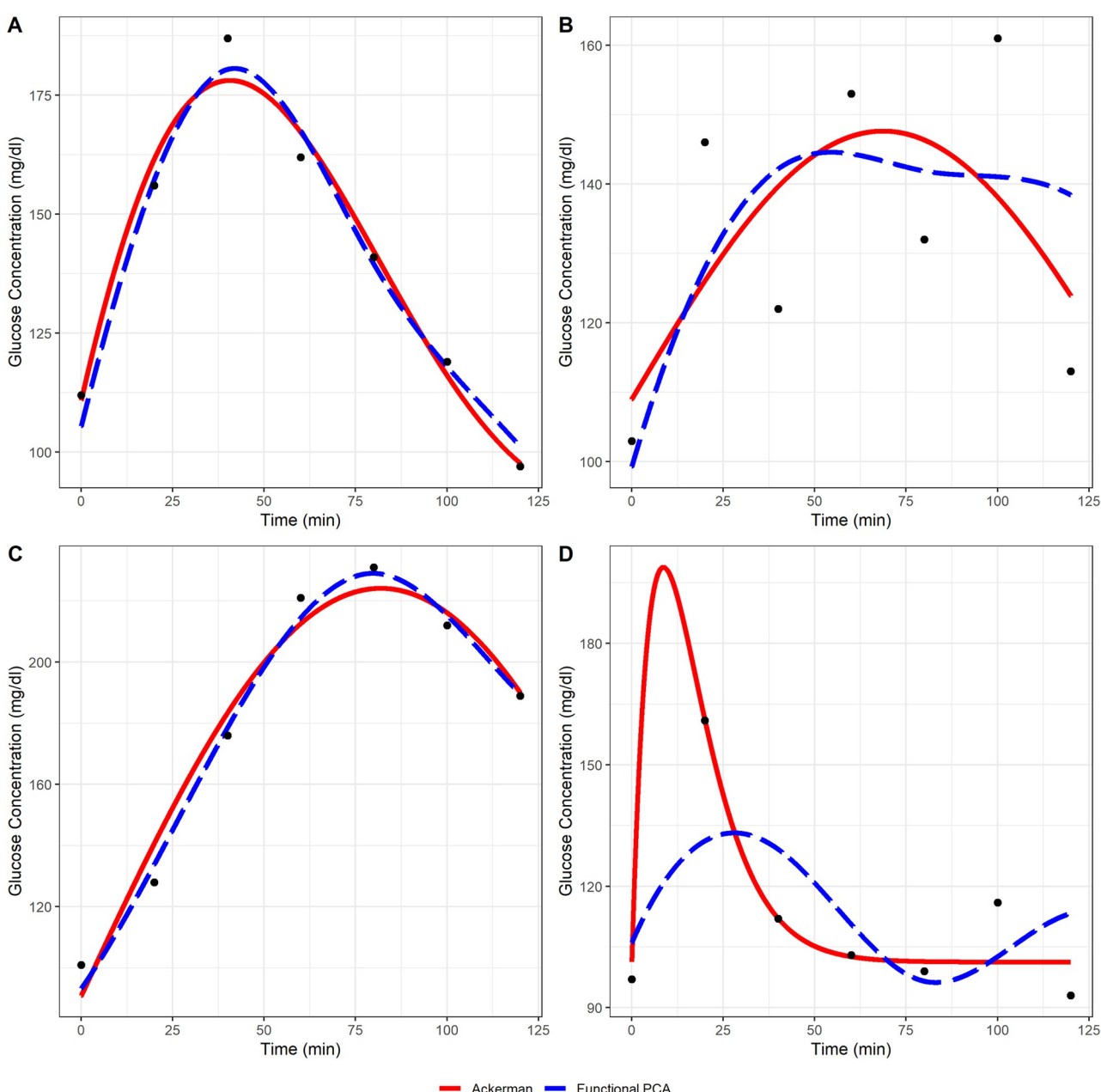

**Fig 6. Predicted OGTT curves from Ackerman and fPCA fits against observed data.** Observed data are shown as black points, the Ackerman model fit as a solid red line, and the fPCA fit as a blue dashed line. Panel A—Adequate Ackerman model fit. Predicted Ackerman and fPCA fit largely agree. Panel B—Abnormal OGTT curve which neither the Ackerman nor fPCA fit model closely. Panel C—Boundary Ackerman model fit ($\hat{k} = 0$). Ackerman and fPCA fit model the observed data closely. Panel D—Extrapolated max Ackerman model fit. The Ackerman fit models the observed data more closely, but the fPCA fit appears more reasonable.

age, and 26% (95% CI: 1%—46%, $P$ = 0.046) lower for White individuals than Black individuals.

Table 2 shows the relationship between estimated effective period and demographic variables for individuals whose OGTT curves were able to be adequately fit by the Ackerman model. Larger effective periods are associated with older age, 0.37 hours per 10 year age

**Table 1. Logistic regression of ackerman model inadequate fit on demographic variables.** Reference group: female sex, Black race, never-smoker.

| Variable | Log Odds Ratio | 95% CI | P-value |
|---|---|---|---|
| Age (10 year change) | 0.15 | (0.02, 0.27) | 0.020* |
| BMI (5 unit change) | −0.03 | (−0.17, 0.11) | 0.687 |
| Sex—Male | −0.24 | (−0.50, 0.01) | 0.056 |
| Race—White | −0.31 | (−0.61, −0.01) | 0.046* |
| Smoking History—Ever Smoker | ∼0 | (−0.26, 0.25) | 0.969 |

**Table 2. Linear regression of estimated effective period on demographic variables.** Adequate fits only. Effective period measured in hours. Reference group: female sex, Black race, never-smoker.

| Variable | Coefficient | 95% CI | P-value |
|---|---|---|---|
| Age (10 year change) | 0.37 | (0.26, 0.49) | <0.001* |
| BMI (5 unit change) | 0.42 | (0.29, 0.56) | <0.001* |
| Sex—Male | 0.34 | (0.10, 0.57) | 0.005* |
| Race—White | −0.01 | (−0.30, 0.28) | 0.955 |
| Smoking History—Ever Smoker | −0.01 | (−0.25, 0.22) | 0.908 |

increase, (95% CI: 0.26—0.49, $P < 0.001$); higher BMI, 0.42 hours per 5 unit increase, (95% CI: 0.29—0.56, $P < 0.001$); and male sex, 0.34 hours greater on average for males than females, (95% CI: 0.10—0.57, $P = 0.005$). These results are in accordance with the hypothesis that longer effective period is associated with poorer glucose regulation, as these demographic variables are known to be associated with poorer glucose regulation.

The associations between the functional principal components and demographic variables are shown in Table 3. PC1 scores are most strongly associated with the demographic variables. PC1 scores are positively correlated with age and BMI, and are higher on average for White individuals than Black individuals. PC2 is negatively correlated with age, is higher in White

**Table 3. Linear regressions of standardized principal components on demographic variables.** Reference group: female sex, Black race, never-smoker.

| Variable | Outcome | Coefficient | 95% CI | P-value |
|---|---|---|---|---|
| Age (10 year change) | PC1 | 0.12 | (0.06, 0.17) | <0.001* |
|  | PC2 | −0.17 | (−0.22, −0.11) | <0.001* |
|  | PC3 | 0.03 | (−0.03, 0.09) | 0.372 |
| BMI (5 unit change) | PC1 | 0.31 | (0.25, 0.37) | 0.001* |
|  | PC2 | ∼0 | (−0.07, 0.06) | 0.948 |
|  | PC3 | 0.03 | (−0.04, 0.10) | 0.349 |
| Sex—Male | PC1 | 0.23 | (0.11, 0.34) | <0.001* |
|  | PC2 | 0.10 | (−0.01, 0.22) | 0.080 |
|  | PC3 | −0.12 | (−0.24, 0.00) | 0.043* |
| Race—White | PC1 | 0.14 | (0.00, 0.28) | 0.049* |
|  | PC2 | 0.41 | (0.27, 0.55) | <0.001* |
|  | PC3 | −0.07 | (−0.22, 0.07) | 0.335 |
| Smoking History—Ever-Smoker | PC1 | 0.07 | (−0.04, 0.18) | 0.263 |
|  | PC2 | 0.13 | (0.02, 0.25) | 0.027* |
|  | PC3 | 0.09 | (−0.03, 0.21) | 0.141 |

individuals than Black individuals, and is higher in ever-smokers than never-smokers. PC3 is lower in males than females. These results show that PC1 is associated with demographic variables that are associated with poorer glucose regulation, and PC2 is associated with demographic variables associated with improved glucose regulation. These results match with the eigenfunctions shown in Fig 4 which indicate PC1 is associated with elevated glucose concentration late into the OGTT, and PC2 is associated with elevated early and reduced late glucose concentrations.

A motivating drive of this research is to test the hypothesis of whether modeling physiological systems provides insights into aging-related outcomes. The chosen outcomes are usual gait speed—one component of Fried's frailty phenotype—and mortality. The purpose of the stage-two models is to test this hypothesis, using OGTT metrics to quantify the functioning of the glucose-insulin system. All of the stage-two models were adjusted for the demographic variables to isolate the contributions of the OGTT metrics. A cubic B-spline with knots at 65, 75, and 85 years was used for age in the gait speed models. For the survival models, increases in age corresponded linearly to increases in the hazard ratio, so a linear age term was used instead of a B-spline.

The Ackerman model has four parameters, $y_F$, $A$, $k$, and $\omega$, two of which combine to form $T_{\text{eff}}$. Rather than using all five quantities as covariates in the stage-two models, the correlations between these parameters were used to select two subsets to avoid issues with multicollinearity. The first subset includes $\hat{Y}_F$, $\hat{k}$, and $\hat{\omega}$; the second includes $\hat{Y}_F$, $\hat{k}$, and $\hat{T}_{\text{eff}}$. For all of the stage-two models with estimated Ackerman model parameters as covariates, a constant term was included to allow for differences in average gait speed between individuals whose Ackerman fit was inadequate or adequate. Additionally, only the estimated Ackerman model parameters from good fits were used in estimating the coefficients for these parameters. This was done by multiplying each estimated Ackerman model parameter term by $\mathbb{1}$(Adequate Fit). This indicator term is 0 for inadequate fits and 1 for adequate fits. This process is equivalent to replacing the estimated Ackerman model parameters for individuals with inadequate fits with the estimated Ackerman model parameter of the adequate fits.

Regression of usual gait speed onto the first set of estimated Ackerman model parameters, $\hat{Y}_F$, $\hat{k}$, and $\hat{\omega}$, and the demographic variables was termed Model 1 and the results are included in Table 4.

Because Ackerman et al. emphasized the importance of $\hat{T}_{\text{eff}}$, the model which includes $\log(\hat{Y}_F)$, $\hat{k}$, and $\hat{T}_{\text{eff}}$ was expanded into three models: one with only $\hat{T}_{\text{eff}}$, one with $\hat{T}_{\text{eff}}$ and $\log(\hat{Y}_F)$, and one including $\hat{T}_{\text{eff}}$, $\log(\hat{Y}_F)$, and $\hat{k}$. These are labeled Models 2–4 and the results for the coefficient for $\hat{T}_{\text{eff}} \cdot \mathbb{1}$(Adequate Fit) are shown in Table 5.

Models 1–4 indicate that none of the estimated Ackerman model parameters are statistically significant predictors of usual gait speed. $\mathbb{1}$(inadequate fit) in Model 1 and $\hat{T}_{\text{eff}}$ in Models 2–4 were the estimated Ackerman parameters most strongly associated with usual gait speed,

**Table 4. Linear regression of usual gait speed (m/s) on standardized estimated ackerman model parameters, model 1.** Adjusted for age, BMI, sex, race, and smoking history.

| Variable | Coefficient | 95% CI | P-value |
|---|---|---|---|
| $\mathbb{1}$(Inadequate Fit) | $-0.02$ | $(-0.05, 0.00)$ | 0.081 |
| $\log(\hat{Y}_F) \cdot \mathbb{1}$(Adequate Fit) | $3.2 \cdot 10^{-3}$ | $(-0.01, 0.02)$ | 0.701 |
| $\hat{k} \cdot \mathbb{1}$(Adequate Fit) | $4.5 \cdot 10^{-3}$ | $(-0.01, 0.02)$ | 0.568 |
| $\hat{\omega} \cdot \mathbb{1}$(Adequate Fit) | $7.7 \cdot 10^{-3}$ | $(-0.01, 0.02)$ | 0.337 |

**Table 5. Linear regression of usual gait speed on standardized estimated ackerman model parameters, models 2–4.** Usual gait speed measured in m/s. Adjusted for age, BMI, sex, race, and smoking history.

| Model, Variable | Coefficient | 95% CI | P-value |
|---|---|---|---|
| Model 2: No additional adjustment<br>$\hat{T}_{\text{eff}} \cdot \mathbb{1}(\text{Adequate Fit})$ | −0.01 | (−0.03, 0.00) | 0.093 |
| Model 3: Adjusted for: $\log(\hat{Y}_F) \cdot \mathbb{1}(\text{Adequate Fit})$<br>$\hat{T}_{\text{eff}} \cdot \mathbb{1}(\text{Adequate Fit})$ | −0.01 | (−0.03, 0.00) | 0.077 |
| Model 4: Adjusted for: $\log(\hat{Y}_F) \cdot \mathbb{1}(\text{Adequate Fit}), \hat{k} \cdot \mathbb{1}(\text{Adequate Fit})$<br>$\hat{T}_{\text{eff}} \cdot \mathbb{1}(\text{Adequate Fit})$ | −0.01 | (−0.03, 0.00) | 0.107 |

both having a negative association with usual gait speed. However, these results indicate the Ackerman model does not appear to be useful for connecting OGTT curves to gait speed.

A linear regression of usual gait speed on the principal component scores and demographic variables was also performed, and the results are shown in Table 6. PC2 was the only principal component score significantly associated with usual gait speed after adjusting for demographic variables. PC2 was found to be positively associated with usual gait speed.

A Cox proportional hazards model including the demographic variables, $\mathbb{1}(\text{Adequate Fit})$, and $\hat{T}_{\text{eff}} \cdot \mathbb{1}(\text{Adequate Fit})$ was constructed to assess whether the Ackerman model is useful for relating OGTT curves to risk of death. The model results are shown in Table 7. Neither Ackerman model term was a significant predictor of mortality.

Lastly, a Cox proportional hazards model including the demographic variables and the principal component scores was fitted. The model results are included in Table 8. PC2 was significantly associated with a reduction in the hazard of death, with a one standard deviation

**Table 6. Linear regression of usual gait speed on standardized functional principal component scores.** Usual gait speed measured in m/s. Adjusted for age, BMI, sex, race, and smoking history.

| Variable | Coefficient | 95% CI | P-value |
|---|---|---|---|
| PC1 | $-6.1 \cdot 10^{-3}$ | (−0.02, 0.01) | 0.348 |
| PC2 | $18.1 \cdot 10^{-3}$ | (0.01, 0.03) | 0.004* |
| PC3 | $-6.7 \cdot 10^{-3}$ | (−0.02, 0.01) | 0.275 |

**Table 7. Cox proportional hazards model of death, standardized estimated ackerman parameters.** Adjusted for age, BMI, sex, race, and smoking history.

| Variable | Hazard Ratio | 95% CI | P-value |
|---|---|---|---|
| $\mathbb{1}(\text{inadequate fit})$ | 1.25 | (0.90, 1.73) | 0.179 |
| $\hat{T}_{\text{eff}} \cdot \mathbb{1}(\text{Adequate Fit})$ | 1.07 | (0.89, 1.31) | 0.467 |

**Table 8. Cox proportional hazards model of death, standardized functional principal components.** Adjusted for age, BMI, sex, race, and smoking history.

| Variable | Hazard Ratio | 95% CI | P-value |
|---|---|---|---|
| PC1 | 0.99 | (0.84, 1.17) | 0.950 |
| PC2 | 0.80 | (0.67, 0.94) | 0.007* |
| PC3 | 1.06 | (0.91, 1.24) | 0.460 |

increase in PC2 score corresponding to a 20% decrease in the expected hazard, holding all else constant.

## Discussion

We encountered challenges fitting a nonlinear model of glucose-insulin dynamics in a cohort of heterogeneous older adults. We identified specific patterns of OGTT curves that the Ackerman model could not adequately fit. These patterns include OGTT curves which are best fit by Ackerman model parameters corresponding to physiologically implausible glucose-insulin dynamics, instances where the likelihood of the Ackerman model is nearly flat, and cases for which the Ackerman model provides poor predictions. Using these patterns, we developed recommendations for classifying dynamical systems model fits as inadequate. For the Ackerman model, these recommendations are to classify as inadequate: model fits with estimated parameters on the boundary of the parameter space, model fits with relatively large differences in the observed and estimated maximum glucose concentrations, and model fits which do not substantially fit the data better than an intercept-only model. Using these criteria, we found the Ackerman model inadequately fits 36% of the observed OGTT curves. Vargas et al. 2022, an arXiv preprint, also developed criteria for identifying inadequate Ackerman model fits. Their study population consisted of 1,911 patients at Mexico General Hospital, ranging from 18 to 80 years old. They employed a Bayesian modeling approach to estimate Ackerman model parameters for each participant. Their model fit inadequately for 32% of the patients' OGTT curves [41]. The similarity in the inadequate fit rates between their study and ours is striking given the differences in the model-fitting algorithms, classification criteria, and study populations. This similarity suggests that our inability to find adequate Ackerman model fits for some OGTT curves is not due to deficiencies in our model-fitting algorithm. While one could suspect OGTT measurement issues as a reason for some of the inadequate fits, it is highly unlikely that a similar extent of issues would be present in two independent studies. Thus, we do not believe that the relatively high percent of inadequate fits is a result of issues with the BLSA's OGTT data in particular.

We also fit the OGTT data using fPCA. The first three eigenfunctions explained over 99% of the variability in the set of smoothed OGTT curves. This indicates that the variability between the predicted OGTT curves using fPCA and the observed OGTT curves is primarily due to the data smoothing process—not due to dropping higher order eigenfunctions. The first eigenfunction appeared to be associated with elevated glucose concentration throughout the OGTT. The second eigenfunction appeared to be associated with elevated glucose concentration near the start of the test and diminished glucose concentration near the end. The third eigenfunction was associated with a bimodal-shaped OGTT curve.

We did not find that the Ackerman model summaries of OGTT curves were significantly associated with usual gait speed or hazard of death when controlling for demographic variables. Other studies have provided evidence that OGTT curves provide information relevant to survival probability [11, 13–16]. Others have shown that there is a relationship between individuals with high fasting glucose levels and slowed gait speed and lower extremity function [42–44]. There is reason to believe the same for gait speed; the lack of association in these analyses with the estimated Ackerman model parameters may be due to the inability of the Ackerman model to capture the relevant information contained in the OGTT curves. This could be a result of the Ackerman model oversimplifying the underlying physiology, especially for late-middle and older adults.

The Ackerman model is a simplified representation of the glucose-insulin system. Two of the most impactful assumptions are that insulin and other hormones are treated identically

and the that rate of change in absorption of the glucose load from the intestines can be modeled as a Dirac $\delta$ function at time 0 [21]. The former simplification assumes that all pertinent regulatory hormones, such as insulin, glucagon, GLP-1, and amylin, can be modeled as one in their effect on glucose concentration. Because these hormones are also time-varying and are not perfectly synchronized, this simplification means the Ackerman model cannot model changes in glucose concentration due to changes in the concentrations of these hormones relative to each other in time. A more accurate model would individually model these hormones, or allow for time-varying parameters due to differences in secretion rates. However, both of these approaches would further complicate parameter estimation.

The assumption that the rate of change in absorption of the glucose load from the intestines can be modeled as a Dirac $\delta$ function at time 0 means the Ackerman model cannot properly model heterogeneity in glucose absorption. Because the intestinal absorption function affects the entire duration of the modeled OGTT curve, many of the OGTT curves the Ackerman model failed to adequately fit may have been due to issues with this assumption. Carbohydrate absorption rate is affected by sex and height [45]. Gastric emptying has been shown to be one of the main sources of variability in OGTT curves repeated on the same individuals, and gastric emptying varies between individuals with normal and impaired glucose-insulin systems [46]. Differences in gastric emptying rates have been estimated to account for approximately 34% of the variability in peak glucose concentration among individuals without diabetes [47]. Dumping syndrome, where food moves rapidly from the stomach to the small intestine, often induces an exaggerated insulin response and hypoglycemia during an OGTT [48]. Some models of blood glucose concentration during an OGTT, such as the oral minimal model, propose explicitly modeling the intestinal absorption term [19]. Of course, this comes with the disadvantage of needing to estimate additional parameters—requiring additional OGTT timepoints.

The simplifications the Ackerman model assumes can result in model fits that inadequately model observed OGTT curves. In some cases, the best-fitting Ackerman model may severely overestimate or underestimate the peak glucose concentration, as with extrapolated fits. In cases where the underlying dynamics are vastly different than what the Ackerman model predicts, the Ackerman model fit produces a low $R^2_{\text{pseudo}}$. Other times, the best-fitting Ackerman model lies in the restricted parameter space, producing a boundary fit. In all of these cases, the estimated Ackerman model parameters cannot be used in the stage-two analyses as the estimates are unreliable. This reduction in sample size hinders our ability to detect associations between estimated Ackerman model parameters and gait speed or mortality. Even worse, the missingness induced by excluding the estimated Ackerman model parameters for inadequate fits is highly related to the shape of the OGTT curve and the associated individual's demographic variables. By the construction of the boundary fit criterion, individuals with low $k$ and $\omega$, and consequently large $T_{\text{eff}}$, parameters are more likely to be excluded from the stage-two analyses than individuals with more average parameters. Excluding this subgroup of individuals also decreases power by reducing the variance in the estimated Ackerman model parameters used in the stage-two analyses. However, the relationship between the ability of the model-fitting algorithm to find an adequate Ackerman model fit and these outcomes are not straightforward, as the inadequate fit intercept term was not significant in any model. This indicates that the individuals with OGTT curves that could not be adequately fit by the Ackerman model are a heterogeneous group and are not uniformly slower or more at risk of death.

In addition to the inadequate fits in which the Ackerman model fails to provide a biologically plausible fit or cannot reasonably model the observed data, the Ackerman model can also be too flexible for the observed OGTT curves. Relatively large changes in small $\hat{\omega}_i$ values can

be counteracted by large changes in $\hat{A}_i$ values to produce nearly identical model fits. This phenomenon can also occur for other parameter combinations and does not necessarily result in a boundary fit classification. Because these fits produce nearly identical OGTT curve predictions, there is large uncertainty in the estimated Ackerman model parameters that should be propagated to the stage-two model. The relatively large number of parameters that must be estimated for the Ackerman model (four) compared to the number of observed data points for each OGTT curve (seven), is a major contributor to these estimation issues. Alternative parametric models, such as the oral minimal model, contain even more parameters that must be estimated. Many of these parametric models were created with the assumption that blood glucose concentrations would be sampled more frequently throughout the course of the OGTT. The original study proposing the oral minimal model, for example, included 21 timepoint samples during the OGTT [19]. The seven measurements captured in the BLSA's OGTT protocol are insufficient to fit these parametric models. Seven measurements are already more than many studies collect, and in practice, only two or three timepoints are often collected. Based on these realities, and the difficulties experienced even with seven time points, the Ackerman model may not be practical for fitting OGTT curves in a clinical setting to make assessments of aging trajectories.

For this research, fPC score estimates were used from all OGTT curves in the stage-two models. A sensitivity analysis was conducted by fitting the stage-two models using only the fPC scores corresponding to the OGTT curves which the Ackerman model fit adequately. These models did not show any significant deviations from the models fit using all OGTT curves. It appears that fPCA is not as sensitive to outlier OGTT curves as the Ackerman model is.

The fPCA results show, as expected, that the OGTT curves do contain information relevant to gait speed and survival, independently from demographic variables. Individuals with greater PC2 scores showed increased gait speed and survival on average than counterparts with lower PC2 scores. These associations indicate that individuals who exhibit earlier peak glucose concentration and diminished glucose concentration after 80 minutes compared to the average have greater gait speed and survival on average after adjusting for demographic variables. Higher PC2 scores indicate a glucose-insulin system that is more capable of responding to and recovering from a glucose stimulus. Although PC1 explains more variance in the OGTT curves about the mean curve than PC2, individuals' PC1 scores were not associated with gait speed nor survival. While this may seem counterintuitive, similar results have been found in prior research. Frøslie et al. found that only PC2 scores differed between women with and without gestational diabetes [34]. Ramsay et al. 2005 noted that functional principal component curves in general follow sinusoidal patterns, and that higher-order components tend to have higher frequency component curves [35].

One of the primary aims of this research was to develop a robust process for fitting a range of OGTT curves that could be encountered in practice. We set the age range for our study to include late middle aged adults (50–64 year olds) rather than strictly older adults (aged 65 years or more) to include more OGTT curves that could potentially challenge our fitting process, as well as to increase our sample size. Because we included late middle aged adults and BLSA participants are relatively healthy compared to similar-age individuals from the US population, we did not use incident frailty in our regression analyses. Instead, usual gait speed and mortality were selected.

This paper has several strengths. It was the first to explore associations between a parametric model of blood glucose concentration during an OGTT, gait speed, and mortality in a population of adults. The BLSA's OGTT data is rich, containing seven time points per OGTT.

This paper provides a case study of the challenges of fitting a nonlinear model to real-world, discretely-sampled data. Two rounds of models were fit: stage-one models, including the parametric Ackerman model and nonparametric fPCA, were fit to OGTT curves; stage-two models, including linear regression and Cox proportional hazards, modeled aging-related outcomes using the estimates produced in stage-one models. The aging outcomes, usual gait speed and mortality, were only significantly associated with the fPCA summaries of the OGTT curves. Parametric models appear to poorly model OGTT curves when the blood glucose concentration is not frequently sampled. fPCA appears to capture information from OGTT curves relevant to aging outcomes.

This paper also has some weaknesses. The Ackerman model-fitting algorithm was only able to adequately fit 64% of the observed OGTT curves. The upper bounds imposed on the Ackerman model parameters were selected to ensure model convergence and with the understanding that extreme parameter values correspond to unrealistic biological phenomena, but the exact chosen bounds are not well supported. fPCA was performed on the entire set of OGTT curves, so data abnormalities or outlier curves may impact the calculated fPC scores. The uncertainty around the parameter estimates in the stage-one models was not carried through to the stage-two models; this was not deemed to be a concern as the Ackerman model parameters were not statistically significantly associated with the outcomes in the stage-two models. For researchers using a similar methodology, this uncertainty would need to be propagated to properly assess the standard errors of the parameter estimates in the stage-two models.

In future work, we plan to compare the use of Ackerman model parameter estimates and fPC scores to commonly-used clinical OGTT measures such as FPG, 2hPG, and HbA1C, and to OGTT composite measures commonly used in research such as area-under-the-curve (AUC) and Matsuda index for predicting aging outcomes. Our hypothesis is that fPC scores will prove to be most strongly correlated with frailty and survival as they capture information about both the magnitude of glucose exposure and the shape of the glucose curve, whereas commonly-used composite measures focus on the former and shape-based summaries on the latter. Expanding on our methodology, we plan to more accurately fit the Ackerman model parameters by modeling the insulin response solution to the Ackerman model differential equations using OGTT insulin data. Additionally, we plan to assess whether alternative OGTT designs can provide better estimates of the Ackerman model. In particular, we suspect a longer test which includes a blood draw at three hours post-load would aid in assessing individuals with slow recoveries. Lastly, we plan to obtain additional BLSA data to assess whether these OGTT curve metrics are predictive of frailty incidence.

## Supporting information

**S1 Fig. Histograms of $R^2_{\text{pseudo}}$ of Ackerman model fits.** Red vertical line indicates $R^2_{\text{pseudo}} = 0.7$. Panel A—Ackerman model fits from all OGTT curves. Panel B—Ackerman model fits excluding boundary fits.
(TIF)

**S2 Fig. Scatterplot matrix of estimated Ackerman model parameters and fPC scores including all OGTT curves.**
(TIF)

**S3 Fig. Residuals boxplots from Ackerman model and fPCA fits for all OGTT curves.** Outliers above 20 mg/dl and below −20 mg/dl were excluded from the plot but were included in constructing the boxes.
(TIF)

## Acknowledgments

We thank the National Institutes of Health and BLSA participants for providing the data used in this study.

## Author Contributions

**Conceptualization:** Grant Schumock, Karen Bandeen-Roche, Rita R. Kalyani, Ravi Varadhan.

**Data curation:** Chee W. Chia, Luigi Ferrucci.

**Formal analysis:** Grant Schumock, Karen Bandeen-Roche, Ravi Varadhan.

**Funding acquisition:** Karen Bandeen-Roche, Chee W. Chia, Luigi Ferrucci, Ravi Varadhan.

**Methodology:** Grant Schumock, Karen Bandeen-Roche, Ravi Varadhan.

**Project administration:** Grant Schumock, Karen Bandeen-Roche, Ravi Varadhan.

**Software:** Grant Schumock, Ravi Varadhan.

**Supervision:** Karen Bandeen-Roche, Ravi Varadhan.

**Validation:** Grant Schumock, Karen Bandeen-Roche, Chee W. Chia, Rita R. Kalyani, Luigi Ferrucci, Ravi Varadhan.

**Visualization:** Grant Schumock.

**Writing – original draft:** Grant Schumock, Karen Bandeen-Roche, Ravi Varadhan.

**Writing – review & editing:** Grant Schumock, Karen Bandeen-Roche, Chee W. Chia, Rita R. Kalyani, Luigi Ferrucci, Ravi Varadhan.

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
