## [Decision Letter · Decision Letter 0]

5 Jan 2024

PONE-D-23-29929Nonlinear modeling of oral glucose tolerance test response to evaluate associations with aging outcomesPLOS ONE

Dear Dr. Schumock,

Thank you for submitting your manuscript to PLOS ONE. After careful consideration, we feel that it has merit but does not fully meet PLOS ONE’s publication criteria as it currently stands. Therefore, we invite you to submit a revised version of the manuscript that addresses the points raised during the review process.

We look forward to receiving your revised manuscript.

Kind regards,

Burak Bayraktar

Academic Editor

PLOS ONE

Journal Requirements:

Additional Editor Comments :

“The BLSA continuously enrolls healthy adults aged 20 years and older. Participants undergo extensive, 3-day testing every 1-4 years, with older participants visiting more frequently. Participants provide written informed consent at each visit.” What is the content of BLSA? For readers who don't know, tell us about it in a few sentences.

Reviewers' comments:

Reviewer's Responses to Questions

**Comments to the Author**

1. Is the manuscript technically sound, and do the data support the conclusions?

Reviewer #1: Partly

Reviewer #2: Yes

2. Has the statistical analysis been performed appropriately and rigorously? 

Reviewer #1: Yes

Reviewer #2: I Don't Know

3. Have the authors made all data underlying the findings in their manuscript fully available?

Reviewer #1: No

Reviewer #2: Yes

4. Is the manuscript presented in an intelligible fashion and written in standard English?

Reviewer #1: Yes

Reviewer #2: Yes

5. Review Comments to the Author

Reviewer #1: The authors present an application of a personalisable parametric-model and fPCA to quantify variation in OGTT glucose response trajectories, and use these reduced dimensional representations of the meal response curve to relate glucose and insulin dysregulation to measures of frailty, namely gait speed and mortality, in a population of 1120 adults. The authors identify some challenges in generating personalised parametric-models, and propose a number of recommendations for evaluating personalised model fits. I find the approaches presented in this manuscript to be highly innovative and believe the open and thorough discussion of the limitations of current glucose-insulin models in the context of generating personalised models to be very timely and relevant for the advancement of the field. I do however have a few points that I would like some clarification on.

For the description of the Ackerman model, what do the parameters in the model (l1-l6) represent. I could also not relate the formulation of the Ackerman model presented to in this to the equations presented in the references 1965 paper, namely how l5 and I related to the K term presented in that paper. Secondly, are equations 1 and 2 printed correctly? I do not understand why I, which is reported to describe the rate of appearance of glucose from the gut, negatively contributed to the rate of change of plasma glucose.

The authors report that the model is fully identifiable. Firstly, what do the authors mean by identifiable? Do they refer to structural identifiability which is a feature of the model structure itself or practical identifiability which characterised the ability to estimate unique parameters given the available data? How was this model identifiability determined? The authors later report that several models are classed as inadequate due to “near redundant” fits, which to me sounds like an identifiability issue. Was practical identifiability assessed for each individual or for the population average?

How many models are lost/deemed inadequate due to each of the three criteria outlined in the results section, namely parameters hitting boundaries, extrapolated fits, and pseudo-R-squared? How does this compare to the fPCA approach, are any individuals lost in this approach?

My understanding is that Y_Fi corresponds to the basal glucose level in the plasms, I also understand that this parameter is being estimated from data in this study. Would it not be possible to directly infer a value for this parameter based on the fasting glucose value or could potentially using some form of regularization aid with the identification of an identifiable parameter set for all individuals? Do the authors have specific reasons to not use the measured fasting glucose concentration as a proxy for Y_Fi?

The authors report using the nls function in R, from their description I assume this is a local least squares solver where parameter fitting was run until a ‘satisfactory’ fit was obtained. What is the distribution of the number of initialization required before an acceptable model fit is found? Did the authors to any testing to evaluate if a global optimum had been obtain? Moreover, related to parameter identifiability, was there any test performed to ensure the model had converge to a unique local minimum? Namly, were any checks in place to demonstrate that the best fitting model had been obtained, or just that an acceptable model fit have been achieved. What might the implication be for the interpretability of the parameter estimates?

With regard to the extrapolated fits, the authors report that model fits where absolute difference between the maximum predicted glucose concentration and observed glucose concentration exceed a specified threshold. However, if I understand correctly the OGTT procedure makes use of discrete sampling (0,20,40,60,80,100,and 120 minutes) is it not possible that the true glucose peak occurs between two sampling points and consequently many be higher than what has been observed? May this exclusion threshold be too stringent?

Reviewer #2: It's a very interesting article to read. It has very well described the difficulties of using the Ackerman model. The content of the paper should be reviewed by a biostatistician. In this study, the authors analyzed data obtained during 75-gram oral-glucose tolerance tests (OGTT) on 1,120 adults older than 50 years of age from the Baltimore

Longitudinal Study on Aging.

The biostatistician can say if their conclusion is valid on the sample size they used.

They adopted a two-stage modeling -First, they fitted OGTT curves with the Ackerman model—a nonlinear, parametric model of the glucose-insulin system—and with functional principal components analysis.

Authors then fitted linear and Cox proportional hazards models and evaluated whether usual gait speed and survival are associated with the stage-one model summaries.

In their conclusion the Ackerman model was unable to adequately fit 36% of the OGTT curves. The stage-two regression analyses found no associations between Ackerman model summaries and usual gait speed, nor with survival. They found functional principal component score was

associated with faster gait speed (p<0.01) and improved survival (p<0.01).

6. PLOS authors have the option to publish the peer review history of their article (what does this mean?). If published, this will include your full peer review and any attached files.

Reviewer #1: No

Reviewer #2: No

---

## [Author Response · Author response to Decision Letter 0]

29 Feb 2024

The Tex file has been added.

The data cannot be made publicly available without restrictions because it would violate the patient consent form, and public availability could compromise patient privacy. Data can be made available for individual studies upon request to the NIH IRB through the BLSA website [https://www.blsa.nih.gov]. This restriction has applied to other papers published with PLOS ONE involving BLSA data (e.g. https://journals.plos.org/plosone/article?id=10.1371/journal.pone.0167241). If our data availability statement is not sufficient, we are happy to modify it if specific wording is required.

Point being addressed 

-Response

*Changes Made

Additional notes

EDITOR: 1. Please ensure that your manuscript meets PLOS ONE's style requirements, including those for file naming. The PLOS ONE style templates can be found at

*1. format changed to double spacing

*2. Fig 5 title altered to include a period at end

*3. Removed spaces from figure files’ names

 If there are any additional formatting requirements we missed, we am happy to update the manuscript again

EDITOR: 2. Note from Emily Chenette, Editor in Chief of PLOS ONE, and Iain Hrynaszkiewicz, Director of Open Research Solutions at PLOS: Did you know that depositing data in a repository is associated with up to a 25% citation advantage (https://doi.org/10.1371/journal.pone.0230416)? If you’ve not already done so, consider depositing your raw data in a repository to ensure your work is read, appreciated and cited by the largest possible audience. You’ll also earn an Accessible Data icon on your published paper if you deposit your data in any participating repository (https://plos.org/open-science/open-data/#accessible-data). -We cannot do this due to restrictions with the BLSA’s IRB agreement

*none 

*1. Removed text “Participants provide written informed consent at each visit. The BLSA protocol has been approved by the Intramural Research Program of the US National Institute on Aging and the Institutional Review Board of the National Institute of Environmental Health Sciences. All data used in this study were fully anonymized by the National Institute on Aging before being provided to the study team.” (lines 92-97)

*2. Added text “Data used in this manuscript are from the Baltimore Longitudinal Study of Aging (BLSA) which is approved by the Institutional Review Board of the National Institute of Environmental Health Sciences under protocol #03-AG-0325. BLSA participants given written informed consent at each study visit. The National Institute of Aging provided the data used in this manuscript to the study team upon review of the study plan and completion of a data use agreement. The data provided by the BLSA to the study team were retrospective and anonymized and thus were acknowledged to not be human subjects research by the study team's IRB, the Johns Hopkins Medicine IRB.” (lines 105 to 115)

EDITOR: 4. We note that you have indicated that there are restrictions to data sharing for this study. PLOS only allows data to be available upon request if there are legal or ethical restrictions on sharing data publicly. For more information on unacceptable data access restrictions, please see http://journals.plos.org/plosone/s/data-availability#loc-unacceptable-data-access-restrictions. 

-We cannot upload our data due to the consent form used in the BLSA. We have modified our Data Availability statement to reflect this.

*New Data Availability statement: This paper relies on data from the Baltimore Longitudinal Study of Aging (BLSA), which are not publicly available due to the consent form language that the BLSA participants agreed to. The National Institutes of Health IRB supervises the BLSA and must approve any data release. To request data from the BLSA, please visit the BLSA website [https://www.blsa.nih.gov]/ and fill out a form. The BLSA Data Sharing Proposal Review Committee, which evaluates and approves all data requests/releases, meets once a month. 

EDITOR: “The BLSA continuously enrolls healthy adults aged 20 years and older. Participants undergo extensive, 3-day testing every 1-4 years, with older participants visiting more frequently. Participants provide written informed consent at each visit.”

What is the content of BLSA? For readers who don't know, tell us about it in a few sentences.

-Thanks for the suggestion. We have added some additional context.

*1. Removed “The BLSA continuously enrolls healthy adults aged 20 years and older.” (lines 90-92)

*2. Added “The BLSA is a community-based cohort study that continuously enrolls healthy volunteers aged 20 years and older who live within two hours driving from Baltimore, Maryland \\cite{Ferrucci2008}. The BLSA follows participants for life. The study was designed to answer mechanistic questions about aging and the transition from health to disease with age.” (lines 98 to 102 )

REVIEWER 2: Frailty: define the term –line23

-Added

*added: “---those demonstrating decreased resilience to stressors, as defined by the Fried frailty index---” (lines 23_ to 24_ ) 

REVIEWER 2: Line49- needs reference

-Added the references from the subsequent two sentences

*Added citations for Bolie1961, Brubaker2007, Man2002, Gatewood1968, Ackerman1964 (line 51 ) 

REVIEWER 2: Line57- what are the 4 parameters added additional text

*Added “---one which estimates fasting glucose, another which controls the rate of an exponential decay function, one which controls the sinusoidal pattern of the glucose curve, and the last influences the amplitudes of the glucose curve oscillations.” (lines _60 to 63_ ) 

REVIEWER 2: Line 63 -vardhan et al which year?

-Modified two sentences to include the year

*Removed “We set out to apply Varadhan et al.'s framework” (lines 37 to 38)

*Added “We applied the framework laid out in Varadhan et al. 2008” (lines 38 to 39 )

and 

*Changed “In applying Varadhan et al.'s framework,” (line 63)

 to

 “In applying the framework described in Varadhan et al. 2008,” (line_68)

REVIEWER 2: Line65-66- reference-“We found greater variability in the shapes of OGTT curves between 65 individuals than expected”. curves between 65 individuals than expected”.

-Modified the text to clarify that we were not comparing the amount of variability that we observed to a known standard

 *Removed “We found greater variability in the shapes of OGTT curves between individuals than expected.” (lines 65-66)

*Added “We found significant variability in the shapes of OGTT curves between individuals.” (lines 70_ to 71 _) 

REVIEWER 2: Line80-Specify “these”.- Does it mean the curves? 

-Modified the sentence to make it more clear 

*Removed “and reports metrics by which to identify these.” (lines 79-80)

*Added “. We termed the Ackerman model fits associated with these curves ``inadequate fits'' and we present the criteria we devised to identify these fits in the results section.”

(lines _86 to 88 _)

REVIEWER 1: For the description of the Ackerman model, what do the parameters in the model (l1-l6) represent. I could also not relate the formulation of the Ackerman model presented to in this to the equations presented in the references 1965 paper, namely how l5 and I related to the K term presented in that paper.

-Added descriptions for l1-l6. In the 1965 paper, they show the equations for the change in glucose and insulin compared to baseline levels. Since l2 and l5 represent average rates of release independent of glucose/insulin (respectively), they drop out. The K term in the 1965 paper is the rate or injection of H per unit blood volume. Since we using OGTT data, there is no insulin injected---that is why we do not have a K term. In other GTT studies, such as IVGTT, K could be some non-zero function.

*Added: “$l_1H$ is the average rate of insulin removal independent of glucose, $l_2$ is the average rate of release of insulin from the pancreas independent of glucose, $l_3Y$ is the net increase in the rate of release of insulin due to glucose, $l_4Y$ is the average rate of glucose removal independent of insulin, $l_5$ is the average rate of release of glucose into the blood, and $l_6H$ is the net increase in the average rate of glucose removal from the blood due to insulin \\cite{Ackerman1964}” (lines 166 to 171) 

REVIEWER 1: Secondly, are equations 1 and 2 printed correctly? I do not understand why I, which is reported to describe the rate of appearance of glucose from the gut, negatively contributed to the rate of change of plasma glucose.

-Thank you for pointing out the mistake in equation 2. It has been corrected. Equation 1 was correct (pg 205 in Ackerman 1964)

*changed “-I” to “+I” in equation 2 

REVIEWER 1: The authors report that the model is fully identifiable. Firstly, what do the authors mean by identifiable? Do they refer to structural identifiability which is a feature of the model structure itself or practical identifiability which characterised the ability to estimate unique parameters given the available data? How was this model identifiability determined? The authors later report that several models are classed as inadequate due to “near redundant” fits, which to me sounds like an identifiability issue. Was practical identifiability assessed for each individual or for the population average?

-Thank you for these excellent points. For that sentence (line 196), we were referring to structural identifiability. We have modified this line to reflect this.

 We did find practical identifiability to be an issue, and we were referring to this with the section discussing “near redundant” fits. We have added text to this section to clarify. Practical identifiability was assessed for each individual using profile likelihoods. We found that for the majority of our OGTT curves, the 

 Ackerman model was practically identifiable. For a subset of the curves, we found that multiple sets of parameters produced equal (or near-equal) fits of the data in terms of likelihoods. The profile likelihoods for this subset revealed nearly flat profile likelihoods near the optimum when profiling over A_i, k_i, and w_i. 

 However, we found this subset of curves was identified as being problematic from our extrapolated fit and boundary fit criteria. Therefore, we are essentially throwing out the parameters estimated from these practically non-identifiable fits before running the stage-two regression models since they are unreliable. I 

 have added some text to describe that we examined profile likelihoods to evaluate practical identifiability.

*removed “fully” line 218

*added “structurally” (line 219)

*added: “These fits are caused by practical non-identifiability.” (lines 351-352)

*added: “In a sensitivity analysis, we created profile likelihood plots for each OGTT to examine for practical non-identifiability of the Ackerman model. We found several instances of practical non-identifiability, as evidenced by flat profile likelihoods near profile maxima. However, these OGTT curves were found to be a 

 subset of the inadequate fits we already identified, so these Ackerman model fits were already excluded from the stage-two models.”

 (lines 363 to 368)

REVIEWER 1: How many models are lost/deemed inadequate due to each of the three criteria outlined in the results section, namely parameters hitting boundaries, extrapolated fits, and pseudo-R-squared? How does this compare to the fPCA approach, are any individuals lost in this approach?

-The number of inadequate fits for each type were/are listed in lines (394 to 396 first submission, 435 to 438 revised submission). This paragraph was not clearly denoted as separate from the section on low-R2_pseudo fits previously, and this has been revised.

 Using fPCA, we did not create a separate classification for predictions which poorly fit the observed data as we did with the Ackerman model. fPCA is more flexible than the Ackerman model, so we did observe any curves as obviously not biologically plausible as we did with the Ackerman model. Furthermore, the fPCs 

 do not have the same clear interpretation as the Ackerman model parameters do (e.g. k_i<0 would indicate increasing oscillations in blood glucose). However, the predicted OGTT curves from fPCA did not always match the data closely. We added text to address your latter question as well as to clarify the meaning of 

 explaining 99% of variability in the smoothed OGTT curves.

*added “the smoothed” (line 442-443)

*added “smoothed” (line 646)

*added “This indicates that the variability between the predicted OGTT curves using fPCA and the observed OGTT curves is primarily due to the data smoothing process---not due to dropping higher order eigenfunctions.” (lines 646 to 649)

*added: “Unlike with the Ackerman model, we did not intentionally identify poor-fitting predicted OGTT curves fit using fPCA. First, the eigenfunctions do not have a direct biological interpretation like the Ackerman model parameters do. While certain combinations of PC scores could lead to predicted OGTT curves which are not biologically plausible, this cannot be determined by examining only one PC score in isolation. Second, each PC score is influenced by all of the other curves in the data set. The Ackerman model is fit independently to each individual's OGTT curve so isolating inadequate fits does not impact other fits.” (lines 467 to 474)

REVIEWER 1: My understanding is that Y_Fi corresponds to the basal glucose level in the plasms, I also understand that this parameter is being estimated from data in this study. Would it not be possible to directly infer a value for this parameter based on the fasting glucose value or could potentially using some form of regularization aid with the identification of an identifiable parameter set for all individuals? Do the authors have specific reasons to not use the measured fasting glucose concentration as a proxy for Y_Fi?

-It would be possible to use the glucose measurement at time 0 directly for Y_Fi rather than estimating it. Then, rather than estimating 4 parameters using 7 data points, we would estimate 3 parameters from 6 data points. By instead estimating Y_Fi, we are in essence doing a regularization by allowing the time points 

 at 20, 40,…,120 minutes influence the value for Y_Fi. In turn, we are able to use the time point at 0 minutes in the estimation of A_i, k_i, and w_i. We experienced the most difficulty with estimating A_i and w_i: A_i and w_i could be varied inversely to provide nearly identical SSRs with significant changes in A_i and 

---

## [Decision Letter · Decision Letter 1]

3 Apr 2024

Nonlinear modeling of oral glucose tolerance test response to evaluate associations with aging outcomes

PONE-D-23-29929R1

Dear Dr. Schumock,

We’re pleased to inform you that your manuscript has been judged scientifically suitable for publication and will be formally accepted for publication once it meets all outstanding technical requirements.

Kind regards,

Burak Bayraktar

Academic Editor

PLOS ONE

Additional Editor Comments (optional):

Reviewers' comments:

Reviewer's Responses to Questions

**Comments to the Author**

1. If the authors have adequately addressed your comments raised in a previous round of review and you feel that this manuscript is now acceptable for publication, you may indicate that here to bypass the “Comments to the Author” section, enter your conflict of interest statement in the “Confidential to Editor” section, and submit your "Accept" recommendation.

Reviewer #1: All comments have been addressed

Reviewer #2: All comments have been addressed

2. Is the manuscript technically sound, and do the data support the conclusions?

Reviewer #1: Yes

Reviewer #2: Yes

3. Has the statistical analysis been performed appropriately and rigorously? 

Reviewer #1: Yes

Reviewer #2: N/A

4. Have the authors made all data underlying the findings in their manuscript fully available?

Reviewer #1: No

Reviewer #2: Yes

5. Is the manuscript presented in an intelligible fashion and written in standard English?

Reviewer #1: Yes

Reviewer #2: Yes

6. Review Comments to the Author

Reviewer #1: Thank you very much for your response, the authors have addressed my previous comments. I find this work on generating personalised computational models to quantify OGTT responses to be very timely and highly relevant.

Reviewer #2: All the questions have been addressed. I think it is a good study where authors have done a two stage model. In the first Ackerman model whether it can determine age related function decline, second they fit linear and Cox proportional hazards models to evaluate gait speed and survival are associated with the stage-one model summaries. From their study the Ackerman model was unable to adequately fit 36% of the OGTT curves. The stage-two regression analyses found no associations between Ackerman model summaries and usual gait speed, nor with survival.

7. PLOS authors have the option to publish the peer review history of their article (what does this mean?). If published, this will include your full peer review and any attached files.

Reviewer #1: No

Reviewer #2: **Yes: **Shashwati Bhattacharya

---

## [Editor Report · Acceptance letter]

29 Apr 2024

PONE-D-23-29929R1 

PLOS ONE

Dear Dr. Schumock, 

I'm pleased to inform you that your manuscript has been deemed suitable for publication in PLOS ONE. Congratulations! Your manuscript is now being handed over to our production team.

Kind regards, 

on behalf of

Dr. Burak Bayraktar 

Academic Editor

PLOS ONE